# Calcium depletion challenges endoplasmic reticulum proteostasis by destabilising BiP-substrate complexes

Steffen Preissler*, Claudia Rato, Yahui Yan, Luke A Perera, Aron Czako, David Ron*

Cambridge Institute for Medical Research, University of Cambridge, Cambridge, United Kingdom

**Abstract** The metazoan endoplasmic reticulum (ER) serves both as a hub for maturation of secreted proteins and as an intracellular calcium storage compartment, facilitating calcium-release-dependent cellular processes. ER calcium depletion robustly activates the unfolded protein response (UPR). However, it is unclear how fluctuations in ER calcium impact organellar proteostasis. Here, we report that calcium selectively affects the dynamics of the abundant metazoan ER Hsp70 chaperone BiP, by enhancing its affinity for ADP. In the calcium-replete ER, ADP rebinding to post-ATP hydrolysis BiP-substrate complexes competes with ATP binding during both spontaneous and co-chaperone-assisted nucleotide exchange, favouring substrate retention. Conversely, in the calcium-depleted ER, relative acceleration of ADP-to-ATP exchange favours substrate release. These findings explain the rapid dissociation of certain substrates from BiP observed in the calcium-depleted ER and suggest a mechanism for tuning ER quality control and coupling UPR activity to signals that mobilise ER calcium in secretory cells.

*For correspondence:
sp693@cam.ac.uk (SP);
dr360@medschl.cam.ac.uk (DR)

## Introduction

The endoplasmic reticulum (ER) of animal cells is distinct in maintaining a high concentration of calcium ($Ca^{2+}$) compared to the surrounding cytoplasm (*Alvarez and Montero, 2002*; *Bygrave and Benedetti, 1996*; *Meldolesi and Pozzan, 1998*; *Solovyova and Verkhratsky, 2002*). Transient $Ca^{2+}$ release from the ER during signalling events controls numerous processes including gene expression and exocytosis (*Carreras-Sureda et al., 2018*; *Raffaello et al., 2016*). It has long been known that pharmacological manipulations (*Macer and Koch, 1988*; *Welch et al., 1983*; *Wong et al., 1993*; *Wu et al., 1981*) and even physiological cues that deplete ER $Ca^{2+}$ (*Sans et al., 2002*) activate transducers of the unfolded protein response (UPR), but the role of proteostatic challenge in UPR activation under these conditions remains unclear.

The folded state of some ER clients, like the low-density lipoprotein receptor (*Fass et al., 1997*), is stabilised by binding $Ca^{2+}$ as a ligand. However, the contribution to UPR activity by the proteostatic challenge arising from the mass of proteins whose maturation is directly compromised by low ER $Ca^{2+}$ (*Cooper et al., 1997*; *Guest et al., 1997*; *Lodish and Kong, 1990*; *Lodish et al., 1992*; *Pena et al., 2010*) is unknown.

Components of the ER protein folding machinery have low-affinity $Ca^{2+}$-binding sites, whose occupancy is likely to decrease upon ER $Ca^{2+}$ depletion within the physiological range. Examples are the lectin-like chaperones calnexin (CNX) and calreticulin (CRT) (*Baksh and Michalak, 1991*; *Wada et al., 1991*), Grp94 (*Macer and Koch, 1988*; *Van et al., 1989*), and protein disulphide isomerase (PDI) (*Lebeche et al., 1994*; *Lucero et al., 1994*; *Macer and Koch, 1988*). These low-affinity binding sites are often found on N- or C-terminal extensions of the aforementioned proteins, a feature consistent with the observation that their core functions as chaperones or oxidoreductases are

not known to be measurably affected at $Ca^{2+}$ levels expected in the depleted ER. In the ER, the functionality of these components might be altered, for example through the effects of changing $[Ca^{2+}]^{ER}$ on their mobility or spatial distribution (*Avezov et al., 2015*; *Corbett et al., 1999*), but presently the significance of this feature remains unknown. Upon $Ca^{2+}$ release from the ER, changes to cytoplasmic $[Ca^{2+}]$ may have indirect effects on the ER environment (*Groenendyk et al., 2014*), but as these are mediated via changes in gene expression, their inherent latency disfavours an important contribution to the rapid post-translational activation of UPR transducers observed in calcium-depleted cells (*Bertolotti et al., 2000*; *Harding et al., 1999*; *Wong et al., 1993*).

Against this background, we were intrigued by the behaviour of the abundant Hsp70-type ER chaperone BiP upon depletion of ER $Ca^{2+}$. Release of ER $Ca^{2+}$ has long been known to correlate with disruption of certain BiP-substrate complexes (*Suzuki et al., 1991*). Such dissociation occurs very rapidly, on the time-scale of minutes (*Preissler et al., 2015*), which is inconsistent with a process driven by the slow build-up of competing unfolded newly synthesised proteins. ER $Ca^{2+}$ depletion is also associated with the appearance of more BiP oligomers (*Preissler et al., 2015*). As chaperoning of unfolded proteins by BiP competes with BiP oligomerisation, the presence of more BiP oligomers is not easy to reconcile with a model whereby a proteostatic challenge arising from the destabilisation of $Ca^{2+}$-binding secreted proteins, or compromised functionality of parallel chaperone systems (e.g. CNX/CRX, PDI), is the major perturbation to ER function imposed by low $[Ca^{2+}]$.

These considerations led us to revisit the effect of ER $Ca^{2+}$ depletion on BiP function in cells and in vitro. Here, we report on a molecular basis of ER $[Ca^{2+}]$-dependent regulation of BiP's functional dynamics under physiological conditions. Our observations provide a plausible explanation for the rapid dissociation of BiP from substrates noted upon ER $Ca^{2+}$ depletion and suggest a chain of causality, whereby a primary effect of $[Ca^{2+}]$ on BiP's chaperone cycle gives rise to a secondary proteostatic challenge in the ER.

## Results

### Divergent effects of ER $Ca^{2+}$ depletion and ER proteostatic challenge on trafficking of a model BiP substrate protein

To gauge the effect of ER $Ca^{2+}$ depletion on trafficking of a model BiP substrate protein through the secretory pathway to the plasma membrane, we chose the T-cell antigen receptor α (TCRα) chain. In absence of other components of the TCR complex, the luminal part of this orphan transmembrane protein interacts with BiP as a substrate and is retained in the ER (*Suzuki et al., 1991*). The reporter consisted of a cleavable, N-terminal ER targeting signal sequence fused to a FLAG-M1 sequence and the ectoplasmic/luminal domain of murine TCRα, followed by a transmembrane domain and a cytosolically localised turquoise fluorescent protein (Tq) (*Figure 1A*). The transmembrane domain of TCRα contains a motif that leads to rapid ER-associated degradation of the orphan subunit (*Bonifacino et al., 1990*). Therefore, it was replaced in this reporter by the transmembrane domain of the interleukin-2 receptor α subunit (*Bonifacino et al., 1991*). We anticipated that reporter transport would expose FLAG-M1 on the cell surface, making it accessible for immunostaining on non-permeabilised cells. Further stringency is imparted by the primary FLAG-M1 antibody that only recognises the epitope on the free N-terminus of TCRα following signal sequence removal. The turquoise fluorescence signal is thus a measure of reporter expression, while the FLAG-M1 signal is specific to the surface-exposed fraction.

Treatment of a CHO-K1 cell line stably expressing the TCRα reporter with thapsigargin (Tg; an agent that causes selective loss of ER $Ca^{2+}$ by inhibiting the ER $Ca^{2+}$ pump SERCA; *Sagara and Inesi, 1991*; *Thastrup et al., 1990*) led to a conspicuous increase of the FLAG-M1 signal, consistent with retention of the reporter in the ER under normal conditions and its $Ca^{2+}$ depletion-induced migration to the cell surface (*Figure 1B*), as previously noted (*Suzuki et al., 1991*).

The TCRα ectoplasmic domain might be subject to multiple competing quality control processes in the ER as it contains several glycosylation sites that may also engage lectin chaperones. To focus the reporter on BiP-mediated regulation, all four glycosylation sites were removed (by single amino acid substitutions), generating a reporter (TCRα-N/Q) that was expressed at lower levels (likely due to enhanced degradation) and was more tightly retained intracellularly under basal conditions and conspicuously exposed on the cell surface upon ER $Ca^{2+}$ release (*Figure 1A and B*).

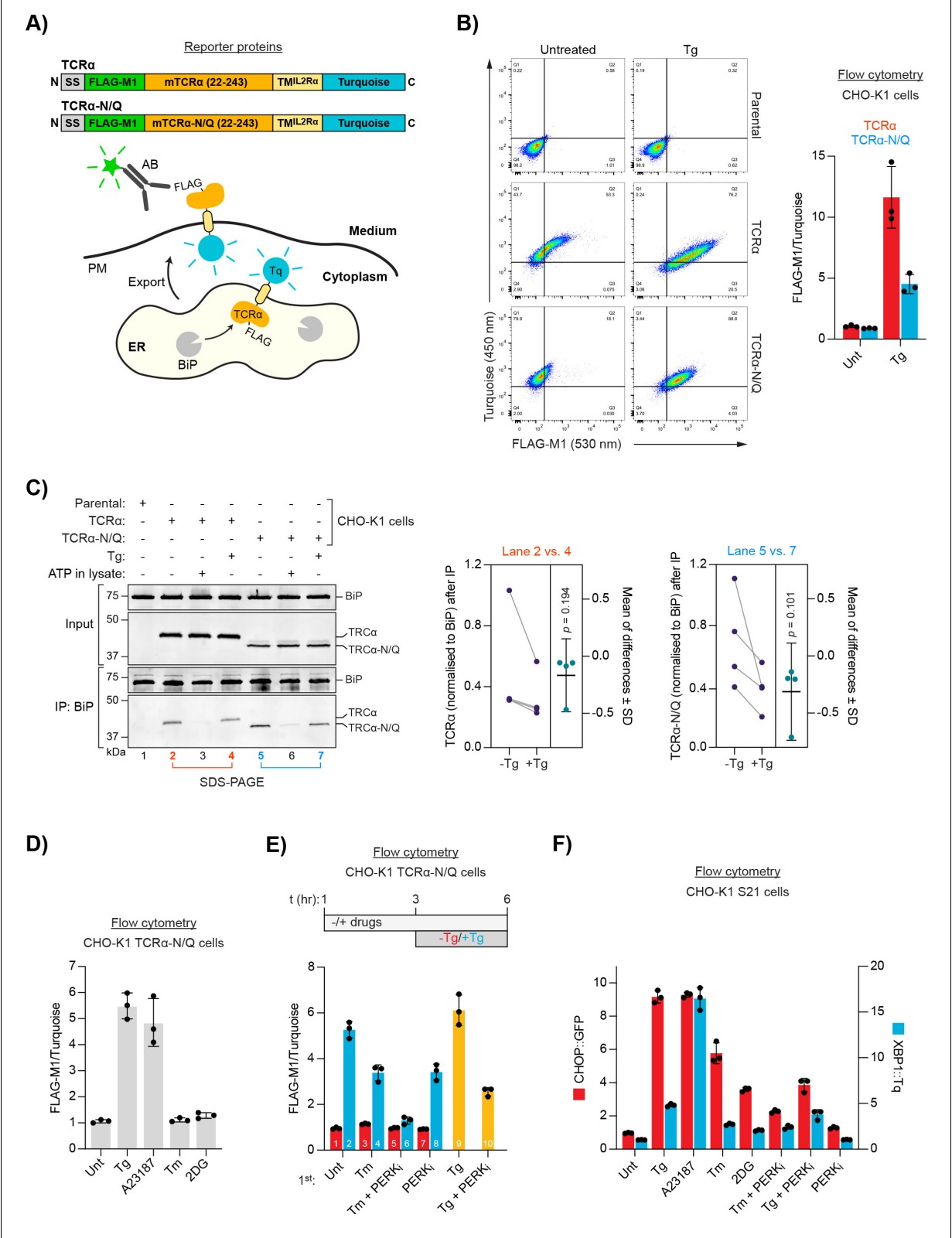

**Figure 1.** ER Ca$^{2+}$ depletion-dependent trafficking of TCRα-based reporter proteins. (**A**) Schematic representation of reporter proteins and detection principles. Signal sequence (SS); FLAG-M1 epitope tag; mouse T-cell antigen receptor α chain (mTCRα); transmembrane domain of the interleukin-2 receptor α subunit (TM$^{IL2Rα}$); monomeric Turquoise (Tq); endoplasmic reticulum (ER); plasma membrane (PM). (**B**) Flow cytometry analysis of TCRα cell surface exposure. CHO-K1 cell lines stably expressing TCRα or glycosylation-deficient TCRα-N/Q were left untreated or exposed to Thapsigargin (Tg) *Figure 1 continued on next page*

Figure 1 continued

for 3 hr before analysis. Dot plots of a representative experiment are shown, along vertical and horizontal guides to facilitate comparisons between the different plots. Graph: reporter surface exposure is plotted as the ratio between the FLAG-M1 and Turquoise median fluorescence signal (derived from all the cells scanned, relative to a data point of TCRα in untreated cells arbitrarily set to 1). Bars represent mean ± SD of the median signal from three independent experiments. *Figure 1—source data 1*. (C) Co-immunoprecipitation (IP) of TCRα reporter proteins with BiP from stable CHO-K1 cell lines (as in 'B') analysed by SDS-PAGE and immunoblotting. Where indicated cells were treated with Tg for 3 hr prior to lysis. ATP was depleted from samples during lysis to stabilise BiP-substrate interactions except from the sample marked with 'ATP' to which additional ATP was added instead. Graph: quantification of TCRα and TCRα-N/Q signals normalised to BiP after IP from untreated (-Tg) and Tg-treated (+Tg) cells and mean of differences ± SD from four independent experiments. A two-tailed, paired, parametric *t*-test was performed. (D–E) Flow cytometry analysis as in 'B'. (D) Cells were treated for 3 hr, as indicated, before analysis [untreated (Unt); tunicamycin (Tm); 2-deoxyglucose (2DG)]. *Figure 1—source data 2*. (E) Cells were exposed to the compounds indicated. After 3 hr, the medium was replaced by medium containing the same compounds without (red) or with additional Tg (blue) for further 3 hr before analysis. Samples 9 and 10 (yellow bars) contained the same amount of Tg before and after medium exchange. PERK inhibitor (PERK$_i$). Bars: mean ± SD from three independent experiments. *Figure 1—source data 3*. (F) Flow cytometry analysis of a UPR reporter CHO-K1 cell line (S21) carrying a predominantly PERK-controlled CHOP::GFP marker and a predominantly IRE1-controlled XBP1::Turquoise marker. The cells were treated for 6 hr, as indicated. Median fluorescence signals relative to untreated cells are shown. Bars: mean ± SD from three independent experiments. *Figure 1—source data 4*.

The online version of this article includes the following source data and figure supplement(s) for figure 1:

Source data 1. Source data for the flow cytometry experiment shown in *Figure 1B*.
Source data 2. Source data for the flow cytometry experiment shown in *Figure 1D*.
Source data 3. Source data for the flow cytometry experiment shown in *Figure 1E*.
Source data 4. Source data for the flow cytometry experiment shown in *Figure 1F*.
Figure supplement 1. Brefeldin A blocks exposure of the T-cell antigen receptor α (TCRα) reporter on the cell surface.
Figure supplement 2. Time course of exposure of the T-cell antigen receptor α (TCRα) reporter on the cell surface during ER Ca$^{2+}$ depletion.
Figure supplement 3. Endoplasmic reticulum (ER) Ca$^{2+}$ depletion and ER stress induce the unfolded protein response (UPR).

Importantly, both wild-type TCRα and TCRα-N/Q co-immunoprecipitated with BiP and the interaction was slightly but consistently diminished in Tg-treated cells (*Figure 1C*). Binding of ATP to BiP lowers its affinity for substrates and enhances their dissociation (*Gaut and Hendershot, 1993*; *Munro and Pelham, 1986*), a feature common to the Hsp70 chaperone family. Addition of ATP to the cell lysates strongly decreased co-precipitation of both reporters with BiP, consistent with their binding to BiP as typical substrates (*Figure 1C*). Although expressed at lower levels, more BiP was bound to TCRα-N/Q compared to the wild-type version, suggesting that the fate of the mutant reporter more strongly depends on its association with BiP (*Figure 1C*, compare lanes 2 and 5). These results agree with a scenario whereby retention of the TCRα proteins in the ER is mediated by their interaction with BiP and Ca$^{2+}$ depletion leads to reporter dissociation from BiP, facilitating ER exit and appearance on the cell surface (*Suzuki et al., 1991*).

This interpretation was further corroborated by the observation that Tg-induced progression of the reporters to the plasma membrane was blocked by brefeldin A (*Figure 1—figure supplement 1*), which disrupts vesicular transport between ER and the Golgi apparatus (*Lippincott-Schwartz et al., 1989*). Moreover, reporter exposure on the cell surface accrued gradually over several hours (*Figure 1—figure supplement 2*), consistent with the kinetics of vesicular transport between ER and the cell surface (*Chen et al., 2013*; *Hirschberg et al., 1998*; *Suzuki et al., 1991*).

Next, we examined whether reporter display on the plasma membrane could also arise from an increased burden of misfolded ER proteins, or if it is a feature specifically related to low [Ca$^{2+}$]$^{ER}$. We used TCRα-N/Q for these experiments due to its improved ER retention characteristics and because it lacks N-linked glycans, rendering the protein indifferent to the direct effects of pharmacological interference with N-linked glycosylation (see below). Depletion of ER Ca$^{2+}$ with the ionophore A23187 also led to accumulation of the reporter on the cell surface (*Figure 1D*). In contrast, the reporter remained intracellular upon treatment with tunicamycin (Tm) or 2-deoxyglucose (2DG) — compounds that cause protein misfolding in the ER by interfering with *N*-glycosylation (*Kozutsumi et al., 1988*; *Kurtoglu et al., 2007*). These observations suggest that release of the reporter's retention by ER Ca$^{2+}$ depletion can be uncoupled from a general proteostatic challenge.

To further address the relationship between proteostatic challenge, ER Ca$^{2+}$ depletion, and TCRα-N/Q trafficking, we temporally separated the perturbation to ER protein folding from ER Ca$^{2+}$ depletion. Treatment of cells with Tm before addition of Tg diminished ER Ca$^{2+}$ depletion-induced reporter exposure on the cell surface (*Figure 1E*, compare bars 2 and 4). Furthermore,

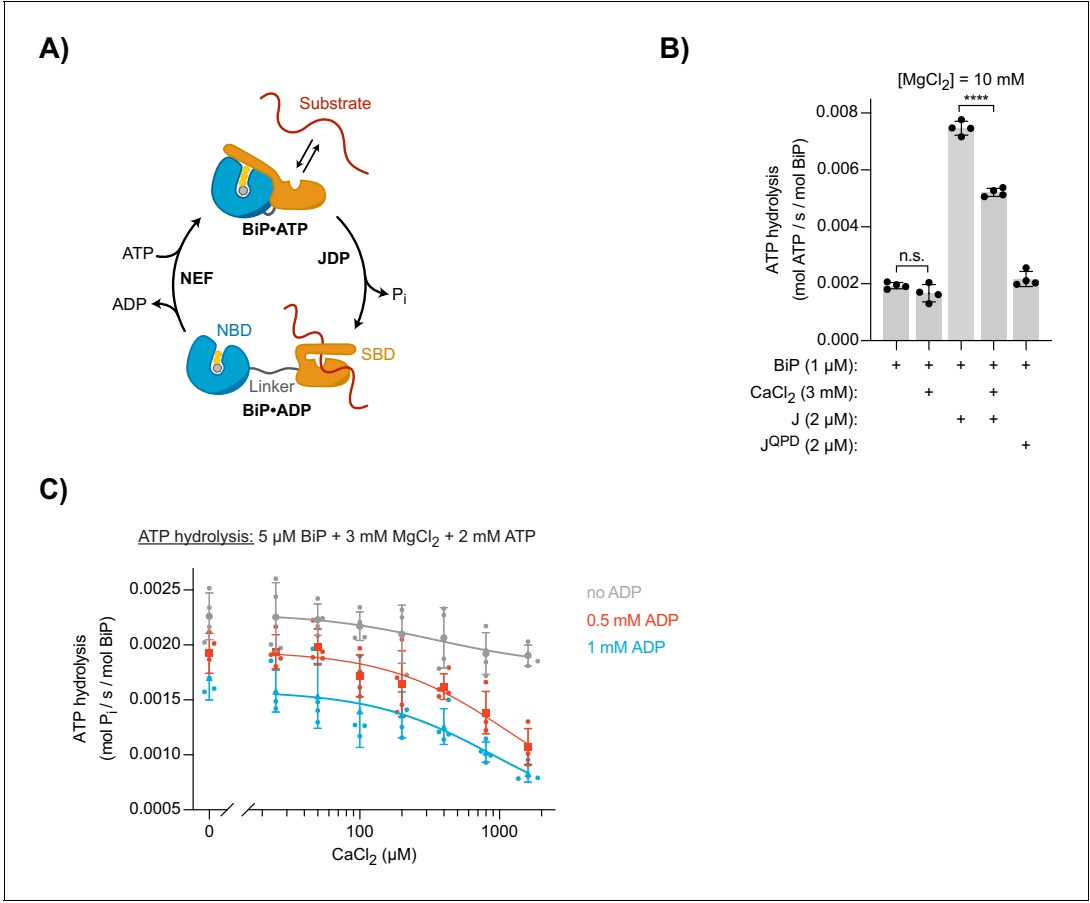

**Figure 2.** Effect of Ca$^{2+}$ on BiP's ATPase activity. (**A**) Schematic representation of BiP's chaperone cycle. Unfolded substrate protein (red); nucleotide binding domain (NBD); substrate binding domain (SBD); interdomain linker (grey); orthophosphate (P$_i$); J-domain protein (JDP); nucleotide exchange factor (NEF). (**B**) ATP hydrolysis by purified BiP measured with a NADH-based ATPase assay. All samples contained 10 mM MgCl$_2$ and 1 µM BiP. Where indicated CaCl$_2$ (3 mM) and wild-type (J) or mutant J-domain (J$^{QPD}$) were added (both at 2 µM). Bars: mean ± SD from four independent experiments. ****p<0.0001, 95% CI −0.002605 to −0.001912; two-tailed, unpaired, parametric *t*-test. *Figure 2—source data 1*. (**C**) ATPase activity of BiP measured by detection of ATP hydrolysis product (P$_i$) accumulation using a malachite green-based assay. All samples contained 3 mM MgCl$_2$ and 2 mM ATP. CaCl$_2$ was titrated (0–1.6 mM). Where indicated samples contained 0.5 mM (red) and 1 mM (blue) ADP. Bold symbols represent mean values ± SD derived from individual data points (staggered dots). Best-fit lines are shown. *Figure 2—source data 2*.

The online version of this article includes the following source data and figure supplement(s) for figure 2:

**Source data 1.** Source data and calculated rates for the ATPase experiment shown in *Figure 2B* and *Figure 2—figure supplement 1*.
**Source data 2.** Source data and calculated rates for the ATPase experiment shown in *Figure 2C*.
**Figure supplement 1.** Effect of Ca$^{2+}$ on BiP's ATPase activity measured by a NADH-based assay.

pharmacological inhibition of basal PERK activity, which increases substrate protein load in the ER by dysregulating the influx of newly synthesised proteins (*Harding et al., 2012*), had a similar effect. The combined application of Tm and PERK inhibitor almost fully blocked subsequent Tg-induced reporter trafficking (*Figure 1E*, bar 6). Thus, ER stress caused by accumulation of unfolded proteins seems to disfavour reporter export upon Ca$^{2+}$ release. However, despite their divergent effects on reporter redistribution, both Ca$^{2+}$-depleting agents and glycosylation inhibitors robustly activated the UPR (*Figure 1F* and *Figure 1—figure supplemnet 3*). These findings hint at a qualitative difference between UPR induction caused by proteostatic challenge and ER Ca$^{2+}$ depletion.

## Ca$^{2+}$ decelerates BiP's ATPase cycle

The dissociation of BiP from model substrates emerges as an intriguing feature that distinguishes ER Ca$^{2+}$ depletion from manipulations that are known to primarily pose a proteostatic challenge. BiP has been shown to bind Ca$^{2+}$ (*Lamb et al., 2006*; *Lièvremont et al., 1997*; *Macer and Koch, 1988*),

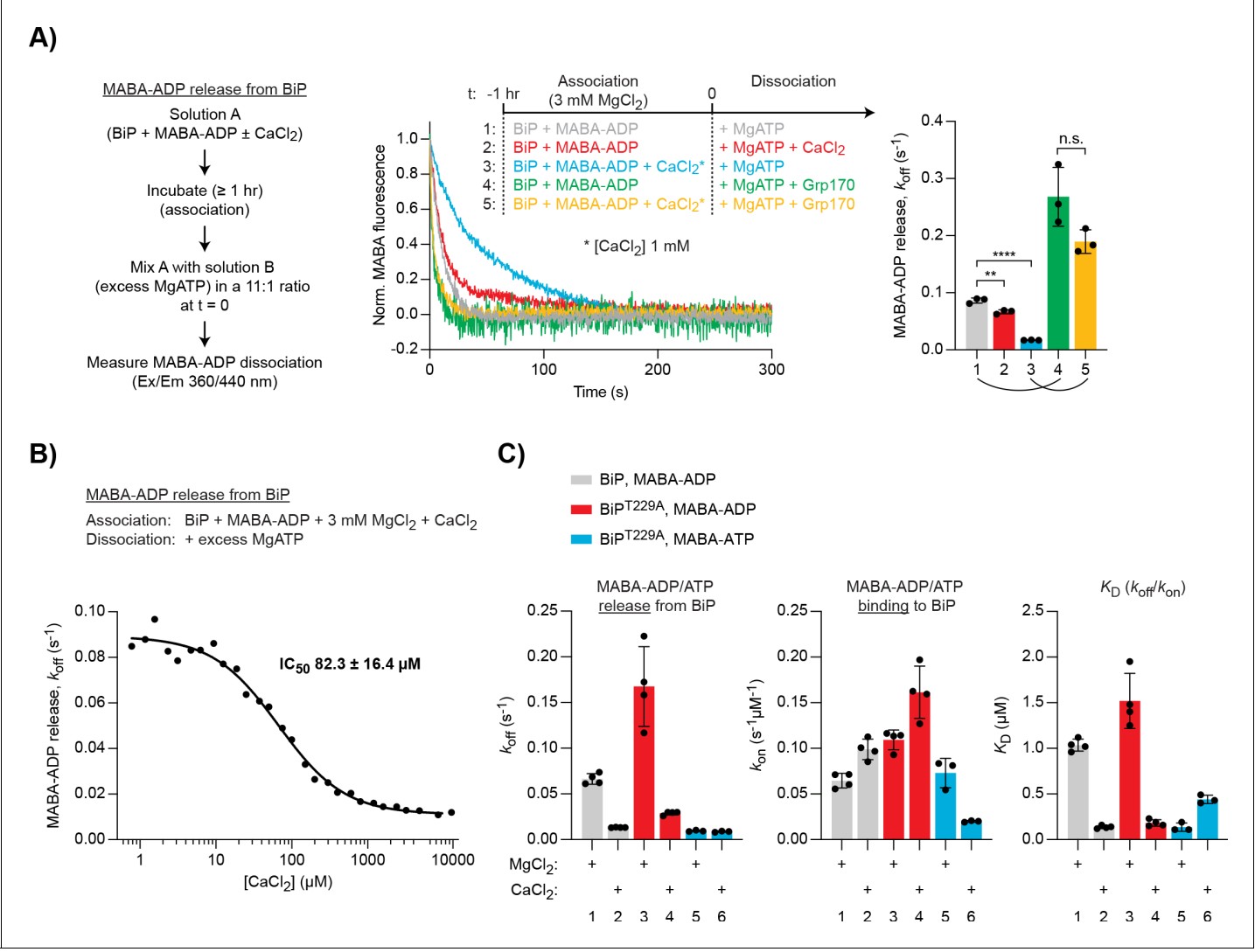

**Figure 3.** Differential effect of Ca²⁺ on ATP and ADP binding to BiP. (**A**) Representative plot of fluorescence against time of pre-formed complexes of MABA-ADP and BiP (each 1.3 μM) challenged at t = 0 with excess of MgATP (1.5 mM) to reveal nucleotide release. All solutions contained 3 mM MgCl₂. CaCl₂ (1 mM) and Grp170 (1.3 μM) were added at t = −1 hr or t = 0 as indicated. Graph: mean MABA-ADP dissociation rate constants ($k_{off}$) ± SD from three independent experiments (the curved lines connecting conditions 1 and 4 as well as 3 and 5 emphasise the effect of Ca²⁺ on the activity of Grp170). *Figure 3—source data 1*. (**B**) MABA-ADP dissociation rates from BiP plotted against [CaCl₂] of a representative experiment. The experiment was performed as in 'A'. All samples contained MgCl₂ (3 mM). CaCl₂ was present at increasing concentrations during nucleotide binding (solution A). The half maximal inhibitory concentration (IC₅₀) of CaCl₂ (mean ± SD) was calculated from three independent experiments. **p=0.0036, 95% CI −0.02899 to −0.01086; ****p<0.0001, 95% CI −0.07659 to −0.06222; two-tailed, unpaired, parametric *t*-test. *Figure 3—source data 2*. (**C**) Effect of Ca²⁺ on affinities of nucleotides for BiP. The dissociation rates of MABA-ADP and MABA-ATP from wildtype or ATPase-impaired T229A mutant BiP were measured as in 'A', whereby either MgCl₂ or CaCl₂ (1 mM) were present throughout the experiment (solutions A and B). The association rate constants ($k_{on}$) were measured upon addition of MABA-labelled nucleotides to BiP in the presence of either divalent cation. The dissociation constants ($K_D$) were calculated based on the rate constants ($k_{off}/k_{on}$). Nucleotide-free proteins were used. Bars: mean ± SD. All the data points from ≥3 independent experiments are shown. Source data and a summary of the calculated values are presented in *Figure 3—source data 3*.

The online version of this article includes the following source data and figure supplement(s) for figure 3:

**Source data 1.** Source data and calculated rates for the MABA-ADP release experiment shown in *Figure 3A*.
**Source data 2.** Source data and calculated rates for the MABA-ADP release experiment shown in *Figure 3B*.
**Source data 3.** Source data and summary of calculated values for the nucleotide binding experiments shown in *Figure 3C*.
**Figure supplement 1.** Dissociation BiP•MABA-ADP complexes formed in the presence of either Mg²⁺ or Ca²⁺.

but a designated $Ca^{2+}$-binding site has not been identified (see below). We therefore set out to investigate the effect of $Ca^{2+}$ on BiP function using cell-free approaches.

Like typical Hsp70s, BiP consists of an N-terminal nucleotide binding domain (NBD) and C-terminal substrate binding domain (SBD) connected by a hydrophobic linker peptide (**Figure 2A**). The binding of BiP to its substrates is subject to an ATP binding- and hydrolysis-dependent allosteric chaperone cycle. In the ATP-bound state, the NBD and SBD are docked against each other and the SBD adopts an open conformation, allowing fast exchange of substrates. ATP hydrolysis to ADP causes domain undocking and closure of the SBD, leading to tight substrate binding. Nucleotide exchange resets the cycle and triggers substrate release. Under physiological conditions (i.e. in excess of ATP), ATP hydrolysis is considered the rate-limiting step of the cycle due to BiP's low intrinsic ATPase activity. However, ER-resident J-domain proteins (JDP) enhance BiP's ATP hydrolysis rate, whereas nucleotide exchange factors (NEF) stimulate ADP release (**Behnke et al., 2015**).

ATP turnover is a global measure of BiP's activity and is reflected in rates of ADP production. A kinetic assay detecting oxidation of nicotinamide adenine dinucleotide (NADH) that is enzymatically coupled to ADP production, confirmed BiP's known low basal ATPase activity and its stimulation by wild-type JDP cofactor, whereas a non-functional version, JDP$^{QPD}$, carrying a H-to-Q mutation in the conserved HPD motif of the J-domain (**Wall et al., 1994**), did not stimulate BiP's ATPase activity (**Figure 2B**). Reaction velocity scaled with BiP concentration, confirming that the assays were performed under non-saturating conditions (**Figure 2—figure supplement 1**). Addition of $CaCl_2$ (to reactions containing magnesium; $Mg^{2+}$) did not significantly alter BiP's basal ATPase rate but inhibited its JDP-stimulated activity by ~30% (**Figure 2B**). The latter may be explained by competition between $Ca^{2+}$ and $Mg^{2+}$ for nucleotides — a CaATP pool is more likely to expose its competitive effect on productive MgATP binding to BiP at JDP-accelerated ATPase rates. This is consistent with the requirement of $Mg^{2+}$ for BiP's ATPase activity (**Kassenbrock and Kelly, 1989**).

These observations seemingly disagree with earlier reports of inhibition of BiP's basal ATPase activity by $Ca^{2+}$ (**Kassenbrock and Kelly, 1989**; **Wei and Hendershot, 1995**). Such differences may relate to the fact that ADP was allowed to accumulate in the assays performed previously, whereas here it was continuously depleted (and ATP regenerated) by the coupled NADH oxidation. However, it is plausible that the presence of ADP might be physiologically relevant and thus important to expose the effect of $Ca^{2+}$ on BiP's function observed in vivo. Corroborating this hypothesis, it has been shown that ADP and $Ca^{2+}$ may modulate each other's affinity for BiP (**Lamb et al., 2006**). Furthermore, recent reports suggest physiological fluctuations in ER luminal nucleotide composition, with lower ATP concentrations in the ER compared with the mitochondria and the cytoplasm (**Depaoli et al., 2018**; **Vishnu et al., 2014**; **Yong et al., 2019**), hinting at a variable and relatively low luminal ATP:ADP ratio.

We therefore turned to an assay that detects orthophosphate production and allows measurement of ATPase activity in the presence of ADP. The basal ATPase activity of BiP (at 3 mM $MgCl_2$ and 2 mM ATP) was only slightly affected when up to 1.6 mM $CaCl_2$ was added (**Figure 2C**). However, in the presence of 0.5 mM ADP ([ATP]/[ADP]=4) titration of $CaCl_2$ showed a clear inhibitory effect, lowering the average ATPase rate of BiP by ~44% at the highest [$CaCl_2$]. $Ca^{2+}$-dependent inhibition was slightly enhanced at 1 mM ADP ([ATP]/[ADP]=2), with a reduction by ~51%. This was observed in the context of generally lower ATPase rates measured in the presence of ADP, which directly competes with ATP for binding to BiP. The dependence of $Ca^{2+}$-mediated inhibition on ADP suggested an important effect of $Ca^{2+}$ on the nucleotide exchange phase of BiP's ATPase cycle.

## $Ca^{2+}$ favours binding of ADP over ATP to BiP

Based on the above observations, we investigated the effect of $Ca^{2+}$ on ADP release from BiP — the first step of nucleotide exchange (**Figure 2A**). Complexes between BiP and MABA-ADP (an ADP analogue whose fluorescence intensity is enhanced when bound to Hsp70s; **Theyssen et al., 1996**), were allowed to form, and the decrease in fluorescence as MABA-ADP dissociated (in the presence of an excess of competing non-fluorescent nucleotides) was measured. In the presence of 3 mM $MgCl_2$, the basal rate of MABA-ADP release from BiP ($k_{off}$ 0.0865 ± 0.0045 s$^{-1}$) was similar to values reported earlier (**Figure 3A**; **Mayer et al., 2003**; **Preissler et al., 2017**). When added during the dissociation phase, $CaCl_2$ (up to 1 mM) had little effect on MABA-ADP release. However, when included alongside the MABA-ADP (and 3 mM $MgCl_2$) during the loading phase, $Ca^{2+}$ strongly

inhibited release in the subsequent dissociation step (decreasing the $k_{off}$ to 0.0171 ± 0.0004 s$^{-1}$) (*Figure 3A*, compare traces 2 and 3).

Importantly, titrating CaCl$_2$ during MABA-ADP binding to BiP (in the presence of 3 mM MgCl$_2$) yielded a half maximal inhibitory concentration (IC$_{50}$) of 82.3 ± 16.4 µM (*Figure 3B*). This value fits well into the estimated range of physiological free Ca$^{2+}$ fluctuations in the ER (from >100 µM to low micromolar concentrations) and suggests that this effect could indeed be relevant to BiP function in vivo.

The NEF, Grp170, enhanced nucleotide release and its net stimulatory effect was considerably greater when BiP•MABA-ADP complexes were formed in the presence of CaCl$_2$ (~3.1-fold [Mg$^{2+}$ alone] vs. ~11.1-fold [Ca$^{2+}$ and Mg$^{2+}$]; *Figure 3A*, compare traces 1 and 4 as well as 3 and 5). As expected, in samples containing only CaCl$_2$, baseline MABA-ADP release was even slower ($k_{off}$ 0.0098 ± 0.0007 s$^{-1}$) and the Grp170-mediated stimulation was enhanced further (~14.7-fold; *Figure 3—figure supplement 1*). The role of NEFs in accelerating ADP release from BiP may therefore be particularly important at the high Ca$^{2+}$ levels of the resting ER.

To explore the affinities of BiP for nucleotides in the presence of Mg$^{2+}$ or Ca$^{2+}$, the association and dissociation rates of MABA-labelled ADP and ATP were compared. Complexes were allowed to form in the presence of either cation. To avoid the confounding effects of ATP hydrolysis, an ATPase-deficient BiP$^{T229A}$ mutant was included (*Gaut and Hendershot, 1993*). In Ca$^{2+}$-containing samples, the dissociation of MABA-ADP from wild-type BiP was again slower, while the association rate was slightly enhanced, resulting in a ~86% lower dissociation constant ($K_D$) for ADP binding to BiP in the presence of Ca$^{2+}$ compared to Mg$^{2+}$ (*Figure 3C*, grey bars). MABA-ADP bound to BiP$^{T229A}$ with a similar $K_D$ (although the rates were somewhat higher), indicating that the T229A mutation has no major impact on BiP's nucleotide-binding properties (*Figure 3C*, red bars). MABA-ATP dissociation from BiP$^{T229A}$ was unaffected by Ca$^{2+}$, but its association was conspicuously slower, leading to a ~3.2-fold higher $K_D$ for MABA-ATP binding to BiP in the presence of Ca$^{2+}$ (*Figure 3C*, blue bars). Thus, Ca$^{2+}$ has opposite effects on the binding of ADP and ATP to BiP: it enhances BiP's affinity for ADP while decreasing the affinity for ATP.

It has been proposed, based on differential scanning and isothermal titration calorimetry data, that Ca$^{2+}$ and Mg$^{2+}$ bind cooperatively with ADP or ATP to BiP (each in a 1:1 stoichiometry) and that both cations occupy the same or overlapping sites in the NBD (*Lamb et al., 2006*). Our findings agree with these conclusions and suggest that Ca$^{2+}$ and Mg$^{2+}$ compete for cooperative binding with either nucleotide to BiP. Moreover, the nucleotide release experiments presented here indicate that Ca$^{2+}$'s effect is played out during BiP•nucleotide complex formation rather than by cation swap on existing BiP•nucleotide complexes. In the presence of large excess of Mg$^{2+}$ and ATP, productive binding of MgATP to Hsp70s is fast and does not limit the rate of re-entering another round of ATP hydrolysis. However, our observations suggest that in the presence of competing ADP and Ca$^{2+}$, cooperative binding of Ca$^{2+}$ with ADP extends the ADP-bound state of BiP during futile ADP-to-ADP exchange, attenuating the chaperone cycle. Increased levels of CaATP, which will also compete with and bind more slowly to BiP than MgATP (*Figure 3C*), may further potentiate BiP exchanging MgADP for CaADP . Thus, at expected physiological ATP:ADP ratios in the ER, Ca$^{2+}$ has a substantial impact on the kinetics of BiP's ATPase cycle and stands to influence its chaperone activity.

## Ca$^{2+}$ stabilises interactions between BiP and substrates

The rate of ADP-to-ATP exchange limits the kinetics with which BiP dissociates from its substrates (*Figure 2A*). This implies that non-productive ADP exchange cycles, favoured by Ca$^{2+}$, might also affect BiP-substrate interactions. We used Bio-Layer interferometry (BLI) to monitor the JDP-mediated binding of BiP to immobilised substrate on the surface of a sensor. This approach tracks transient BiP-substrate interactions in real-time (*Preissler et al., 2017*). A biotinylated J-domain (from ERdJ6) was co-immobilised together with a substrate peptide (P15; *Misselwitz et al., 1998*) on streptavidin-coated BLI sensors (*Figure 4A*). These were then transferred to solutions containing BiP to detect its association, followed by introduction into solutions without BiP to detect its dissociation. The large number and heterogeneity of potential BiP binding sites on protein-coated sensors provides a good means to study the substrate binding characteristics of BiP (*Preissler et al., 2017*). Fast recruitment of BiP was observed in the presence of ATP and Mg$^{2+}$ (*Figure 4A*, traces 1–5). This interaction was dependent on J-domain-stimulated ATP hydrolysis, as the BiP binding signal was weak in the presence of a non-functional J-domain (J$^{QPD}$) (*Figure 4A*, trace six and *Preissler et al.,*

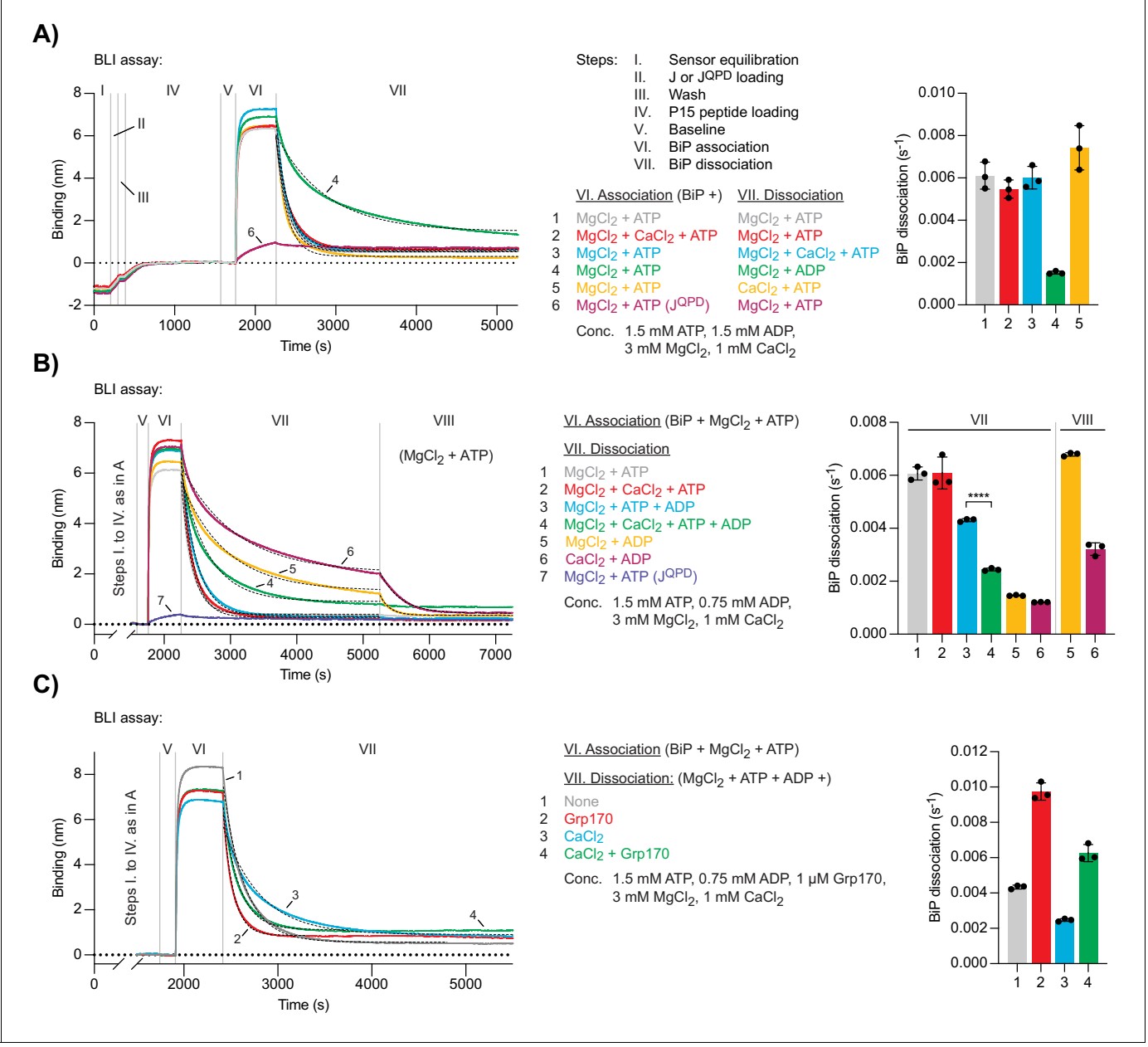

**Figure 4.** $Ca^{2+}$ affects ATP-dependent dissociation of substrates from BiP. (**A**) A representative plot of the Bio-Layer interferometry (BLI) signal against time. The individual steps of the experiment (I-VII) are indicated: following an initial equilibration step (I) biotinylated wild-type or QPD mutant J-domain protein was immobilised corresponding to an interference signal difference of ~0.4 nm (II). After a wash step (III) the sensors were saturated with biotinylated P15 BiP substrate peptide (IV). After a stable baseline signal was established (V) the sensors were transferred into solutions containing BiP to measure association in the presence of ATP (VI). Dissociation of BiP from the sensor was measured in protein-free solutions containing either ADP or ATP (VII). The composition of solutions in steps VI and VII as well as the concentrations of the variable components are indicated. (**A**) $MgCl_2$ and/or $CaCl_2$ were present where indicated. Dashed lines represent single exponential fit curves. Graph: mean dissociation rate constants ± SD from three independent experiments. *Figure 4—source data 1*. (**B**) Same as 'A' but with identical solutions for all sensors in step VI and varying solution compositions in step VII. In the additional step (VIII) the sensors were introduced into solutions containing ATP and $MgCl_2$. ****$p<0.0001$, 95% CI −0.00198822 to −0.00174311, two-tailed, unpaired, parametric *t*-test. *Figure 4—source data 2*. (**C**) Same as 'B' (reactions 3 and 4) with Grp170 present in step VII where indicated. *Figure 4—source data 3*.

The online version of this article includes the following source data for figure 4:

**Source data 1.** Source data and calculated rates for the BLI experiment shown in *Figure 4A*.
**Source data 2.** Source data and calculated rates for the BLI experiment shown in *Figure 4B*.
**Source data 3.** Source data and calculated rates for the BLI experiment shown in *Figure 4C*.

*2017*). Although slightly slower, BiP recruitment was also efficient in the presence of $Ca^{2+}$ ([$MgCl_2$]/[$CaCl_2$]=3), suggesting that $Ca^{2+}$ competes only weakly with $Mg^{2+}$ under these conditions (*Figure 4A*, trace 2); consistent with slower cooperative binding of CaATP to BiP, compared to MgATP, and the absence of an allosteric effect of $Ca^{2+}$ on ATP binding- and hydrolysis-dependent substrate interactions.

BiP molecules bound to the sensor via high-affinity substrate interactions are, by definition, in a post-hydrolysis state — either ADP-bound or nucleotide-free (apo). Thus, BiP dissociation is subordinate to nucleotide exchange events and is accelerated by ATP binding. Accordingly, in $Mg^{2+}$-containing solutions, BiP dissociation was slow in the presence of ADP and fast in the presence of ATP. When present in the dissociation phase, the combination of $Mg^{2+}$ and $Ca^{2+}$ ([$MgCl_2$]/[$CaCl_2$]=3) did not affect the rate of ATP-induced BiP dissociation from sensors (*Figure 4A*, trace 3). Dissociation of BiP was also fast in a solution containing ATP and $Ca^{2+}$ as the only divalent cation, indicating that CaATP is able to allosterically trigger BiP dissociation from its substrates (*Figure 4A*, trace 5). Given that $Ca^{2+}$ slows down the binding of ATP to BiP (*Figure 3C*), it seemed surprising that the average dissociation rate was slightly higher compared to MgATP-induced BiP dissociation. This finding might be explained by poor hydrolysis of CaATP, countering the contribution of ATP-hydrolysis-dependent re-establishment of a stable BiP•substrate complex to the BLI signal arising in the presence of the hydrolysable MgATP.

In the presence of $Mg^{2+}$ and both ATP and ADP ([ATP]/[ADP]=2), the dissociation of BiP from J-domain- and P15-coated BLI sensors was slower compared to a solution containing only $Mg^{2+}$ and ATP, consistent with direct competition of ADP with ATP for binding to BiP (*Figure 4B*, trace 3). Addition of $Ca^{2+}$ ([$MgCl_2$]/[$CaCl_2$]=3) to a solution containing both nucleotides further inhibited dissociation of BiP (*Figure 4B*, trace 4). This result agrees with the notion that $Ca^{2+}$ kinetically promotes the ADP- and substrate-bound state of BiP during a futile nucleotide exchange cycle by decreasing the rate of ADP release from BiP.

To dissect this further, BiP•substrate complexes were assembled on J-domain- and P15-coated BLI sensors (in the presence of MgATP, as above) followed by two sequential dissociation steps. The BiP-loaded sensors were first introduced into solutions containing ADP and either $Mg^{2+}$ or $Ca^{2+}$ to promote formation of BiP•MgADP•substrate or BiP•CaADP•substrate complexes, which represent the expected intermediates of futile nucleotide exchange. The sensors were then transferred to solutions containing $Mg^{2+}$ and ATP. BiP bound to a sensor that had been exposed to $Mg^{2+}$ and ADP dissociated twice as fast in the second step than BiP on a sensor that had been exposed to $Ca^{2+}$ and ADP (*Figure 4B*, compare traces 5 and 6). This observation indicates that the slow release of CaADP kinetically limits ADP-to-ATP exchange and consequently ATP-enhanced dissociation of BiP from substrates. In solutions containing $Mg^{2+}$ and a mixture of ATP and ADP ([ATP]/[ADP]=2), the NEF Grp170 enhanced dissociation of BiP from substrate-loaded BLI sensors both in the absence and presence of $Ca^{2+}$ to a similar extent (on average ~2.2- and ~2.5-fold, respectively; *Figure 4C*). Under these conditions, the stabilising effect of $Ca^{2+}$ on BiP-substrate interactions is thus apparent even when nucleotide exchange is stimulated by NEF. The difference in magnitude of Grp170's relative stimulatory effect on the dissociation of BiP•ADP complexes formed in presence of $Ca^{2+}$ (*Figure 3A*) and resolution of BiP-substrate interactions (with competing ATP/ADP and $Mg^{2+}$/$Ca^{2+}$ being present; *Figure 4C*) may reflect in part the potential for establishment of both BiP•CaATP/BiP•CaADP and BiP•MgATP/BiP•MgADP species during multiple exchange cycles that occur in the BLI setup designed to measure the effects of NEF on BiP-substrate interactions and the single turnover nature of the former assay.

These observations suggest that changes in [$Ca^{2+}$] (within the physiological range) regulate BiP's chaperone activity by modulating the kinetics of productive nucleotide exchange, and thus the rates of substrate release and entry into the next ATPase cycle. These findings propose a mechanism for the decline in BiP-substrate interactions observed upon ER $Ca^{2+}$ depletion. All other things being equal, at high [$Ca^{2+}$]$^{ER}$ (resting state) substrate dissociation is slower. Release of $Ca^{2+}$ from the ER accelerates ATP binding and substrate dissociation, whereas the rate at which ATP hydrolysis-dependent substrate interactions are formed remains largely unaffected. The acceleration of productive nucleotide exchange and substrate dissociation, relative to substrate binding, partitions BiP from its substrates.

## Selective responsiveness of ER-localised Hsp70s to $Ca^{2+}$

$Ca^{2+}$ excursions within the range shown here to affect BiP activity are unique to the ER. Therefore, we deemed it of interest to determine if regulation by $Ca^{2+}$ is a feature selective to ER-localised Hsp70 (i.e. BiP) or common to all Hsp70s, regardless of the $Ca^{2+}$ environment in which they operate. To this end, we produced and purified bacterial DnaK, the *Drosophila melanogaster* homologue of BiP, as well as human cytosolic Hsp70 (HSPA1A) and Hsc70 (HSPA8).

As before, MABA-ADP complexes with the different Hsp70s were formed in the presence of $Mg^{2+}$ and nucleotide release upon exposure to excess of ATP was measured. MABA-ADP release rates varied between the different Hsp70s (*Figure 5A*, dark traces). $Ca^{2+}$ strongly inhibited MABA-ADP release from mammalian BiP and *Drosophila* BiP (by ~79% and ~82%, respectively) but had a more modest inhibitory effect on MABA-ADP release from DnaK (~30%), cytosolic Hsp70 (~37%), and Hsc70 (~50%) (*Figure 5A*). Thus, under these conditions, $Ca^{2+}$ had the strongest inhibitory effect on MABA-ADP release from BiP homologues.

We also analysed JDP-mediated substrate interactions of the different Hsp70 proteins using the BLI assay. Dissociation was performed in the presence of $Mg^{2+}$, ATP, and ADP ([ATP]/[ADP]=2). Further addition of $Ca^{2+}$ had the strongest inhibitory effect on the dissociation of mammalian BiP (~43% decreased dissociation rate) and *Drosophila* BiP (~53% decrease), whereas dissociation of the other chaperones was less affected (Hsp70 ~16%, Hsc70 ~16%, DnaK ~9% decrease; *Figure 5B and C*). These observations suggest that sensitivity of nucleotide exchange and substrate interactions to $Ca^{2+}$ is a feature that is relatively selective to BiP homologues, perhaps having evolved as an adaptation to the ER environment.

## CaADP binding selectively affects BiP's structural stability

To establish if the selective effect of $Ca^{2+}$ on ADP binding to BiP is reflected in the structure of the complex, we purified the NBDs of mammalian BiP, Hsp70, and Hsc70 and co-crystallised them with $Ca^{2+}$ and ADP. High-resolution X-ray crystal structures were obtained for all three NBDs (*Table 1*). In each case, additional density in the nucleotide-binding cleft indicated occupancy by ADP (*Figure 6—figure supplement 1A*) and the overall conformations of the three NBDs were very similar (*Figure 6A*). Overlay of the BiP(NBD) structure with the Hsp70(NBD) and Hsc70(NBD) structures revealed root-mean-squared deviations (RMSD) of 0.769 Å and 0.734 Å over the $C_\alpha$ alignment, respectively. Moreover, in each structure, $Ca^{2+}$ was identified in the nucleotide- binding cleft, where it formed contacts to both phosphate groups (α and β) of the nucleotide as well as indirect contacts to protein residues via several water molecules (*Figure 6A*, inset). The position of the $Ca^{2+}$ ion and the coordinated water molecules were nearly identical in all three complexes and similar to a previously reported lower resolution structure of human BiP bound to CaADP (*Wisniewska et al., 2010*). In contrast, the $Mg^{2+}$ ion in an earlier Hsc70(NBD)•$Mg^{2+}$•ADP structure coordinates only the β-phosphate of ADP directly (*Figure 6B*, upper panel), which may contribute to the lower affinity of MgADP compared to CaADP for NBD binding. The BiP(NBD)•$Ca^{2+}$•ADP complex also resembled the structures of BiP(NBD)$^{apo}$ and the NBD of BiP$^{apo}$ containing both its NBD and SBD (*Figure 6—figure supplement 1B*). The location of $Ca^{2+}$ is slightly shifted with respect to the position of $Mg^{2+}$ in structures that likely reflect the post-hydrolysis state, where the cation coordinates orthophosphate ($P_i$) and the β-phosphate of ADP (*Figure 6B*, lower panel). The latter is more similar to the configuration of Hsp70•MgATP complexes, where the $Mg^{2+}$ is located between the β- and γ-phosphate groups of ATP (*Figure 6—figure supplement 1C*). The $Ca^{2+}$ ion in a structure of human Hsp70(NBD) in complex with $Ca^{2+}$, ADP, and $P_i$ occupies the same position as $Mg^{2+}$ in a post-hydrolysis state NBD structure (*Figure 6—figure supplement 1D*). Together, these findings reveal that $Ca^{2+}$ participates in the coordination of ADP in the NBD, which is consistent with cooperative binding of $Ca^{2+}$ and nucleotide to BiP. The structures also imply that binding of $Ca^{2+}$ and $Mg^{2+}$ is mutually exclusive. This is consistent with evidence against replacement of $Mg^{2+}$ by $Ca^{2+}$ in the post-hydrolysis complex and with evidence that $Ca^{2+}$ binds together with ADP after release of the hydrolysis products (i.e. to the apo NBD) in cycles of ADP rebinding.

The highly similar CaADP binding mode does not explain the stronger effect of $Ca^{2+}$ on BiP function compared to the other Hsp70s. However, the $IC_{50}$ values for $Ca^{2+}$-dependent inhibition of MABA-ADP release from the NBDs of Hsp70 (444.7 ± 80.5 μM) and Hsc70 (345.5 ± 71.1 μM) were much higher than from the NBD of BiP (57.1 ± 7.4 μM) (*Figure 6C*). The latter was similar to the $IC_{50}$

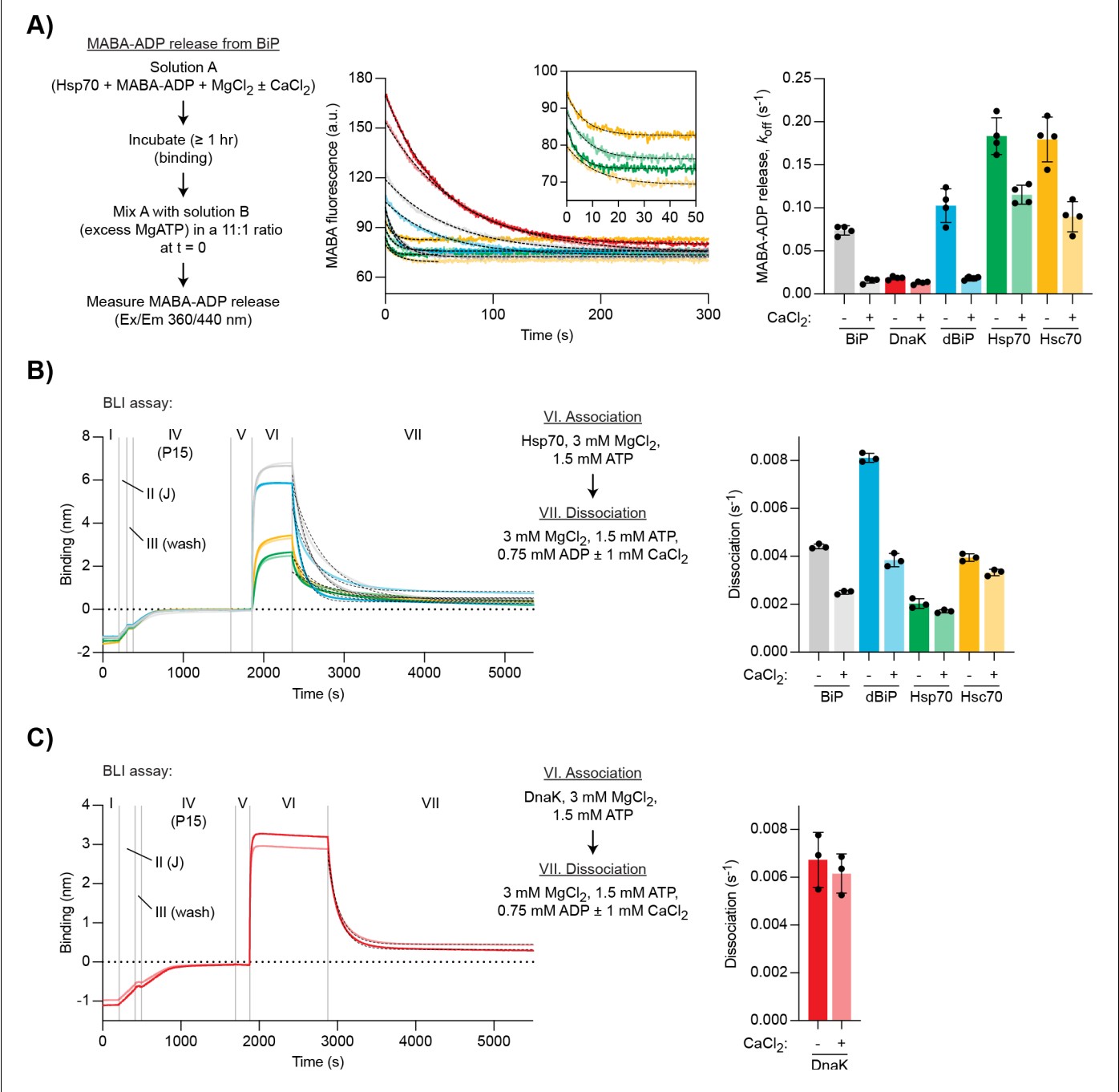

**Figure 5.** Ca$^{2+}$ selectively stabilises BiP•ADP compared with cytoplasmic Hsp70s. (**A**) Representative plot of fluorescence against time of complexes of MABA-ADP and indicated full-length Hsp70 chaperones (each 1.3 μM) formed without or with CaCl$_2$ (1 mM) (dBiP, *Drosophila* BiP). All solutions contained MgCl$_2$ (3 mM). Excess of MgATP (1.5 mM) was added at t = 0 to reveal MABA-ADP release. Final concentrations are given. Graph: mean MABA-ADP dissociation rate constants ± SD from four independent experiments. *Figure 5—source data 1*. (**B–C**) A representative plot of the Bio-Layer interferometry (BLI) signal against time. The experiments were performed as in *Figure 4A* with the indicated Hsp70s. Dissociation of Hsp70s from the sensor was measured in protein-free solutions (VII). The composition of solutions in steps VI and VII as well as the concentrations of the variable components are indicated. (**B**) Analysis of eukaryotic Hsp70s. Dashed lines represent single exponential fit curves. Graph: mean dissociation rate constants ± SD from three independent experiments. *Figure 5—source data 2*. (**C**) Same as '**B**' but with *E. coli* DnaK. *Figure 5—source data 3*. The online version of this article includes the following source data for figure 5:

**Source data 1.** Source data and calculated rates for the MABA-ADP release experiment shown in *Figure 5A*.
**Source data 2.** Source data and calculated rates for the BLI experiment shown in *Figure 5B*.
**Source data 3.** Source data and calculated rates for the BLI experiment shown in *Figure 5C*.

**Table 1.** Data collection and refinement statistics.

| | BiP(NBD) | Hsp70(NBD) | Hsc70(NBD) | Apo BiP oligomer | ADP BiP oligomer |
|---|---|---|---|---|---|
| **Data collection** | | | | | |
| Synchrotron stations (DLS) | I03 | I04 | I04 | I24 | I24 |
| Space group | P1 | $P2_12_12_1$ | $P12_11$ | $P12_11$ | $P12_11$ |
| a,b,c; Å | 48.36, 51.86, 93.08 | 46.17, 63.71, 144.50 | 73.68, 78.04, 75.38 | 50.33, 51.29, 119.72 | 50.18, 51.25, 188.80 |
| $\alpha$, $\beta$, $\gamma$; $^0$ | 78.07, 86.72, 62.28 | 90.00, 90.00, 90.00 | 90.00, 101.26, 90.00 | 90.00, 100.17, 90.00 | 90.00, 99.77, 90.00 |
| Resolution, Å* | 29.14-1.88 (1.92-1.88) | 29.53-1.52 (1.54-1.52) | 73.73-1.85 (1.89-1.85) | 59.92-1.77 (1.81-1.77) | 29.27-1.94 (1.99-1.94) |
| Rmerge* | 0.063 (0.989) | 0.065 (0.703) | 0.120 (1.005) | 0.054 (0.844) | 0.047 (0.533) |
| $<I/\sigma(I)>$* | 18.2 (1.8) | 14.0 (2.3) | 8.9 (1.4) | 9.9 (1.1) | 13.7 (2.0) |
| CC1/2* | 0.999 (0.669) | 0.998 (0.814) | 0.998 (0.633) | 0.975 (0.531) | 0.999 (0.717) |
| No. of unique reflections* | 62200 (3792) | 66932 (3125) | 71566 (4400) | 58872 (3347) | 43705 (2937) |
| Completeness, %* | 97.4 (90.4) | 99.7 (95.1) | 100.0 (100.0) | 99.9 (99.8) | 99.1 (94.5) |
| Redundancy* | 8.9 (7.5) | 6.7 (6.3) | 6.7 (5.7) | 3.2 (3.2) | 3.3 (3.2) |
| **Refinement** | | | | | |
| Rwork/Rfree | 0.209/0.225 | 0.196/0.212 | 0.218/0.247 | 0.185/0.217 | 0.184/0.226 |
| No. of atoms (non H) | 6137 | 3417 | 6399 | 4432 | 4327 |
| Average B-factors | 32.5 | 18.6 | 22.9 | 32.8 | 34.4 |
| RMS bond lengths Å | 0.002 | 0.002 | 0.002 | 0.006 | 0.006 |
| RMS bond angles,$^0$ | 1.163 | 1.163 | 1.172 | 1.341 | 1.338 |
| Ramachandran favoured region, % | 99.34 | 99.17 | 98.28 | 98.66 | 98.27 |
| Ramachandran outliers, % | 0 | 0 | 0 | 0 | 0 |
| MolProbity score[†] | 0.78 (100) | 0.83 (100) | 0.80 (100) | 0.96 (100) | 0.99 (100) |
| PDB code | 6ZYH | 6ZYI | 6ZYJ | 7A4U | 7A4V |

*Values in parentheses are for highest-resolution shell.[†]100[th]percentile is the best among structures of comparable resolutions. 0[th]percentile is the worst.

values of full-length BiP and the *Drosophila* BiP(NBD) (52.4 ± 8.1 µM) (*Figure 3B* and *Figure 6—figure supplement 2A*). In contrast, the $IC_{50}$ for inhibition of MABA-ADP release from DnaK(NBD) by $Ca^{2+}$ was so high (>8 mM) that saturation was not reached over the titration range (*Figure 6—figure supplement 2A*). Given that concentrations of $Ca^{2+}$ in the cytoplasm are in the nanomolar range, these results imply that the effect of $Ca^{2+}$ on ADP binding to cytoplasmic Hsp70s has limited physiological relevance. We also tested the NBD of Kar2, the ER Hsp70 chaperone of yeast (*Saccharomyces cerevisiae*), which had a high $IC_{50}$ of 888.0 ± 233.6 µM (*Figure 6—figure supplement 2A*). Thus, regulation by $Ca^{2+}$ seems specific to metazoan BiP.

The NBD structures also revealed a second, surface-exposed $Ca^{2+}$ ion that is coordinated by the carboxyl group of D257 and backbone carbonyl of H252 in BiP (*Figure 6—figure supplement 3A*) and homologous residues in the cytosolic Hsp70s. This region is implicated as a hinge in the outward movement of NBD domain IIB, that favours bound nucleotide release. Binding of a $Ca^{2+}$ ion in that location may disfavour such movement and stabilise the bound nucleotide. However, the significance of this site to $Ca^{2+}$'s selective effect on BiP was called into question for two reasons: Firstly, this location was only found to be occupied in structures from crystals obtained at particularly high $Ca^{2+}$ concentrations (but not at low millimolar $Ca^{2+}$ concentrations), suggesting low affinity for the cation, and secondly, by the likelihood that this site is accessible to $Ca^{2+}$ also when the BiP NBD is occupied with MgADP — circumstances under which $Ca^{2+}$ does not retard nucleotide exchange (*Figure 3A*). In an effort to resolve this issue experimentally, we mutated D257 in full-length BiP to alanine or asparagine and tested the effect on ADP release. Although the mutations slightly altered the basal and Grp170-stimulated ADP release rates in the presence of only $Mg^{2+}$, both mutants and the wild-type protein showed a similar response to $Ca^{2+}$ when added during BiP•nucleotide complex

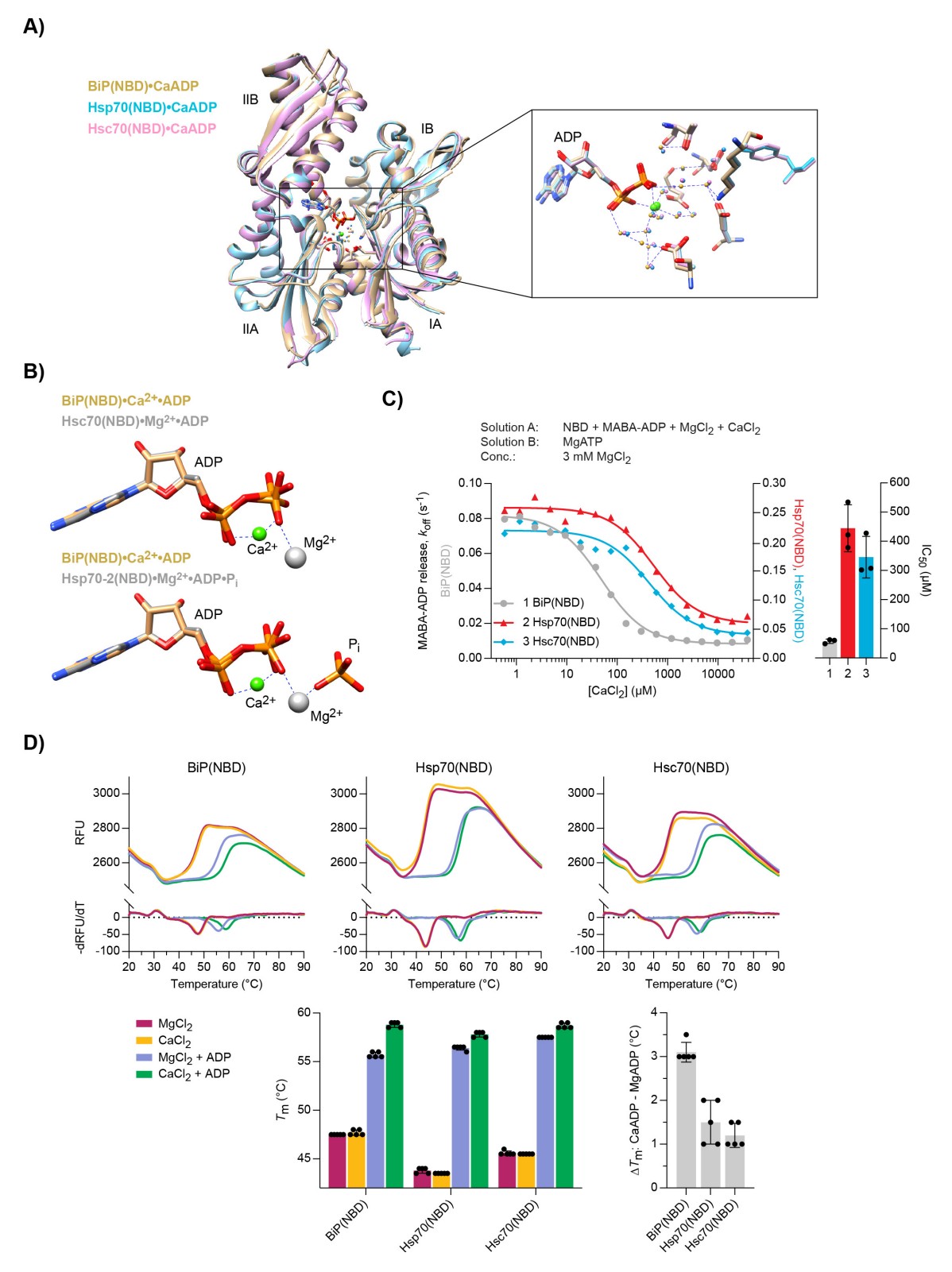

**Figure 6.** Structural analysis of $Ca^{2+}$ and ADP binding to Hsp70 nucleotide binding domains (NBDs). (**A**) Ribbon diagram of the aligned crystal structures of BiP (yellow), Hsp70 (blue), and Hsc70 (pink) NBDs in complex with $Ca^{2+}$ and ADP. The subdomains IA-IIB are indicated. Inset: coordination of $Ca^{2+}$ ions (green) by water (small spheres) and ADP in the nucleotide binding cleft. (**B**) Nucleotide and metal ion ligands of BiP(NBD)•$Ca^{2+}$•ADP (as in 'A') and bovine Hsc70(NBD)•$Mg^{2+}$•ADP (PDB 1BA1) or human Hsp70-2(NBD)•$Mg^{2+}$•ADP•$P_i$ (PDB 3I33) based on nucleotide alignment. ADP, inorganic

*Figure 6 continued on next page*

*Figure 6 continued*

phosphate ($P_i$), $Ca^{2+}$ (green sphere), and $Mg^{2+}$ (grey sphere) are shown. (**C**) MABA-ADP dissociation rates from the indicated NBDs plotted against [$CaCl_2$] of a representative experiment. The experiment was performed as in *Figure 3A*. All samples contained $MgCl_2$ (3 mM). $CaCl_2$ was present at increasing concentrations during NBD•MABA-ADP complex formation. Bars: the half maximal inhibitory concentration ($IC_{50}$) of $CaCl_2$ (mean ± SD) was calculated from three independent experiments. *Figure 6—source data 1*. (**D**) Melting temperatures ($T_m$) of the indicated NBDs (at 5 µM) were measured by differential scanning fluorimetry (DSF) in the presence of $MgCl_2$ or $CaCl_2$ (each 6 mM) without or with ADP (4 mM). Top: representative melt curves with their negative first derivatives (RFU, relative fluorescence units). Left bar graph: mean $T_m$ ± SD of three independent experiments. Right bar graph: difference in $T_m$ ($\Delta T_m$) between MgADP and CaADP containing samples. *Figure 6—source data 2*.

The online version of this article includes the following source data and figure supplement(s) for figure 6:

**Source data 1.** Source data and calculated rates for the MABA-ADP release experiment shown in *Figure 6C*.
**Source data 2.** Source data and calculated melting temperatures for the DSF experiment shown in *Figure 6D*.
**Figure supplement 1.** Structural details of nucleotide binding domains (NBDs) and their ligands.
**Figure supplement 2.** Characterisation of nucleotide binding domains (NBDs) of *Drosophila* BiP, DnaK, and Kar2.
**Figure supplement 3.** The surface-exposed second $Ca^{2+}$- binding site in BiP's NBD does not significantly contribute to the effect of $Ca^{2+}$ on ADP release.

formation (*Figure 6—figure supplement 3B*). These measurements speak against an important role for this second $Ca^{2+}$-binding site in modulating the ADP-binding properties of BiP.

Despite their nearly identical CaADP-bound structures, BiP(NBD), Hsp70(NBD), and Hsc70(NBD) share only 70% sequence identity. We speculated that structural variation that is inapparent in the low-energy, crystallised state may nonetheless lead to functionally important differences in thermo-dynamic stabilisation by $Ca^{2+}$ and ADP. Differential scanning fluorimetry (DSF) measurements revealed that binding of ADP to all NBDs substantially elevated their melting temperatures ($T_m$), as observed previously for full-length BiP (*Lamb et al., 2006*). The $T_m$ increased further in the presence of $Ca^{2+}$ (*Figure 6D* and *Figure 6—figure supplement 2B*). Importantly, the stabilising effect of $Ca^{2+}$ was greatest for mammalian and *Drosophila* BiP(NBD), compared to the cytosolic Hsp70(NBD)s. These findings suggest that the larger decrease in structural flexibility imparted by CaADP binding to the NBD of ER-localised Hsp70s contributes to the relatively stronger inhibition of ADP release by $Ca^{2+}$.

## Differential sensitivity of BiP oligomers and BiP•substrate complexes to $Ca^{2+}$

The activity of BiP in the ER is subject to post-translational regulatory mechanisms, one of which is the transient sequestration of substrate-free BiP into inactive oligomeric assemblies. These homomeric BiP complexes are based on typical substrate interactions between the SBD of one protomer and the interdomain linker of another protomer (*Preissler et al., 2015*). As a consequence, all BiP molecules of an oligomer are in a post-hydrolysis, domain-undocked state typical of ADP-bound or apo Hsp70s (*Figure 7A*). These features entail a chain-like configuration where only the first BiP molecule (A) can bind substrate, whereas the SBDs and linkers of all internal protomers (B) are occupied. Apart from the last molecule (C), which exposes a free linker, all other protomers are conformationally locked and therefore functionally inactive. This model also predicts that oligomer assembly and disassembly occur mainly on one end and involve the 'C' protomer because it provides a binding site for another BiP molecule (in assembly) and is the only protomer able to undergo ATP-induced domain docking (required for dissociation). We define this end of BiP oligomers as the 'plus' (+)-end in analogy to the more dynamic end of cytoskeletal filaments (*Figure 7A*).

The strong propensity of BiP to form oligomers and their architecture is further supported by structural data: despite being C-terminally truncated beyond helix A of the flexible SBD lid structure and containing a substrate binding-weakening V461F mutation, we obtained crystals of BiP in multi-component mixtures formed by substrate interactions between the SBD and the interdomain linker of adjacent molecules in the crystal lattice (*Figure 7B* and *Figure 7—figure supplement 1*).

Due to their characteristics, BiP oligomers are in direct competition with binding of unfolded substrates, which explains their disappearance during ER stress (*Blond-Elguindi et al., 1993*; *Preissler et al., 2015*). This suggests a physiological role of oligomerisation in buffering folding capacity by transient withdrawal of BiP from the pool of active chaperones during periods of declining unfolded protein load. However, the surprising observation that BiP oligomers are enhanced by

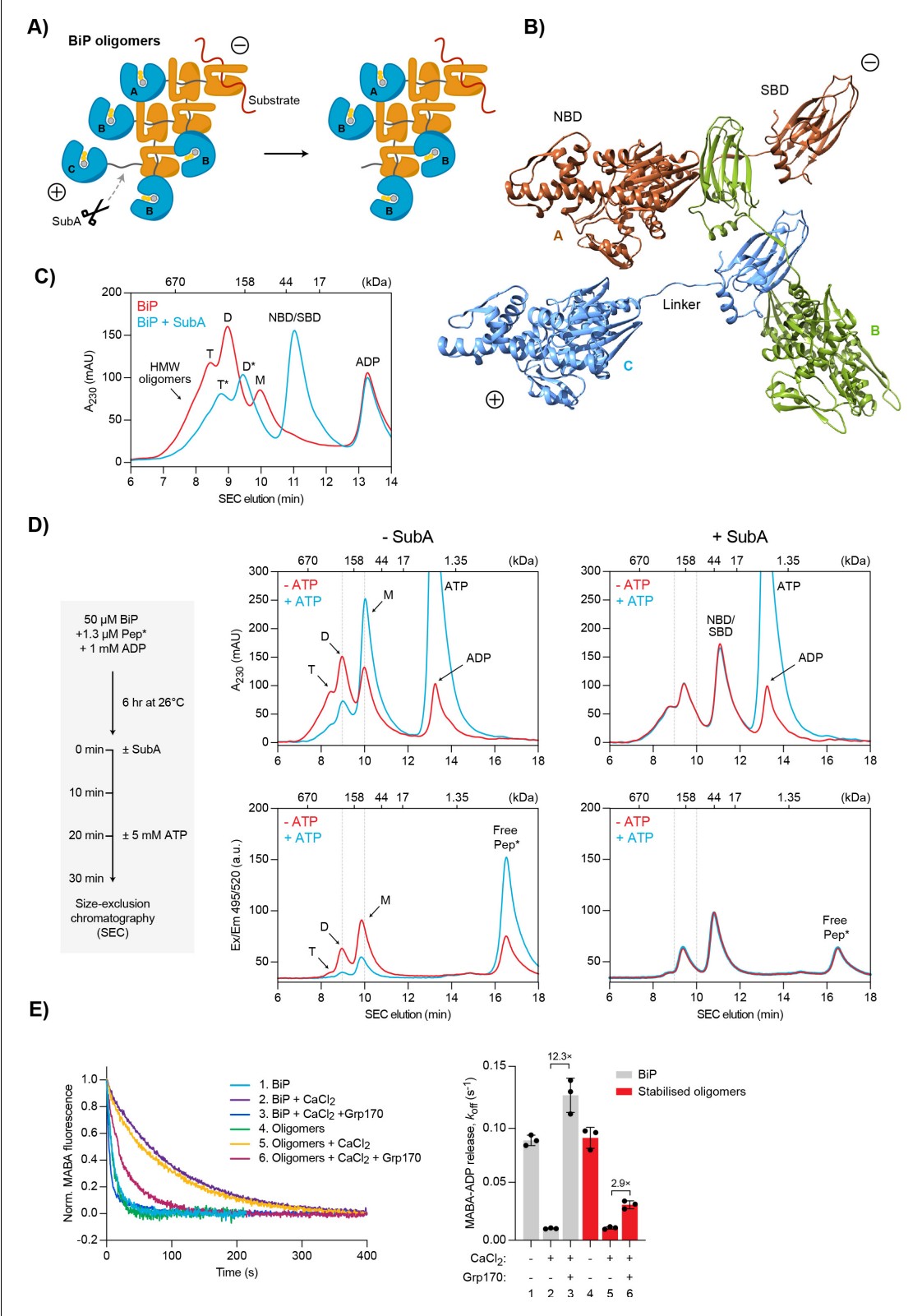

**Figure 7.** Effect of Ca²⁺ on BiP oligomers in vitro. (**A**) Schematic representation of BiP oligomers. Unfolded substrate protein (red) and the distinct types of BiP protomers (A, B, and C) are shown. Oligomers are directional assemblies with minus (-) and plus (+) ends. The cleavage site for the BiP linker-specific protease SubA (scissors) is only accessible on (+)-end protomer (C). The predicted inert BiP oligomer lacking the C protomer nucleotide binding domain (NBD; a result of SubA cleavage) is shown on the right. (**B**) Ribbon diagram of apo BiP^T229A-V461F oligomers formed by typical substrate

*Figure 7 continued on next page*

*Figure 7 continued*

interactions between protomers. Three protomers (labelled A-C according to 'A') are presented in different colours. Note the unusual C- to N-terminal engagement of BiP's interdomain linker as a substrate bound to the substrate binding domain (SBD) of an adjacent BiP molecule. (C) Size-exclusion chromatography (SEC) elution profile of BiP. BiP was incubated with ADP (1 mM) and where indicated treated with SubA before SEC. Monomeric (M), dimeric (D), and trimeric BiP (T) is indicated. Heterogenous high-molecular weight (HMW) BiP complexes elute early as a shoulder. Note that the peaks of proteolytic cleavage products originating from trimers and dimers (*) are shifted to later elution times. Also note the disappearance of BiP monomers and a new peak of the individual NBD and SBD after treatment with SubA. *Figure 7—source data 1*. (D) SEC elution profile as in 'C' of BiP and substrate peptide. BiP was incubated with trace amounts of fluorescently labelled substrate peptide (NR; Pep*) in the presence of ADP at indicated concentrations. Where indicated the samples were exposed at t = 0 to SubA for 30 min before SEC. Where indicated excess of ATP was added 10 min before SEC. Peptide bond absorbance at 230 nm ($A_{230}$) and the fluorescence signal of the labelled peptide were recorded separately. *Figure 7— source data 2*. (E) Representative plot of fluorescence against time of complexes of MABA-ADP and BiP or purified, stabilised BiP oligomers (each 1.3 µM) formed in the presence of either $MgCl_2$ or $CaCl_2$ (each 3 mM). Excess of ATP (1.5 mM) without or with Grp170 (1.3 µM) was added at t = 0 to reveal nucleotide release. Graph: mean MABA-ADP dissociation rate constants ($k_{off}$) ± SD from three independent experiments. Final concentrations are given. *Figure 7—source data 3*.

The online version of this article includes the following source data and figure supplement(s) for figure 7:

**Source data 1.** Source data for the SEC traces shown in *Figure 7C*.
**Source data 2.** Source data for the SEC traces shown in *Figure 7D*.
**Source data 3.** Source data and calculated rates for the MABA-ADP release experiment shown in *Figure 7E*.
**Figure supplement 1.** Structural features of oligomeric BiP.
**Figure supplement 2.** Stability of purified BiP oligomers after cleavage with SubA assessed by size-exclusion chromatography.

ER $Ca^{2+}$ depletion seems to contradict this function and remained unexplained so far (*Preissler et al., 2015*). We revisited this phenomenon in light of the new insights gleaned here.

BiP oligomers rapidly disassemble in vitro in the presence of ATP. However, the hypothesised process by which BiP oligomers form and disassemble, described above, predicts that specific inactivation of the 'C' protomer at the (+)-end would render them refractory to the destabilising effect of ATP. We therefore took advantage of the protease subtilisin A (SubA) that cleaves BiP site-specifically at the linker sequence (*Paton et al., 2006*). Under conditions that favour oligomers, a large fraction of BiP is resistant to cleavage by SubA (*Preissler et al., 2015*), likely because only the linker of the 'C' protomer is accessible to the protease, whereas the linkers of all other protomers are protected by an SBD of another BiP protomer (*Figure 7A*). Indeed, size-exclusion chromatography (SEC) analysis after exposure of BiP to SubA revealed a shift of all peaks to later elution times, consistent with removal of a single NBD from oligomeric structures and complete cleavage of monomers (*Figure 7C*).

We next formed complexes between BiP and fluorescently-labelled substrate peptide by extended incubation of BiP (at concentrations at which it is largely oligomeric) with trace amounts of the peptide. The samples were then analysed by SEC and substrate peptide fluorescence was detected separately from bulk protein absorbance at 230 nm ($A_{230}$; a signal vastly dominated by BiP). The peptide co-eluted with BiP monomers and oligomers (*Figure 7D*, left panels). The stepwise increase in the ratio of $A_{230}$ to fluorescence signal with oligomer size is consistent with each oligomeric species exposing only a single binding site [at its (-)-end]. As expected, addition of ATP to the sample shortly before injection into the SEC column converted BiP oligomers largely into monomers, releasing the bound substrate peptide. As noted previously, treatment with SubA prior to SEC shifted the oligomer peaks to later elution times and abolished the monomer signal in both the $A_{230}$ and fluorescence channel. However, in contrast to the intact oligomers, the cleaved species were completely insensitive to ATP-induced disassembly and peptide release (*Figure 7D*, right panels). We conclude that proteolytic removal of the NBD from the terminal protomer stabilises BiP oligomers by inactivating their only conformationally responsive molecule, the 'C' protomer at the (+)-end. These data also provide evidence for a strictly directional oligomer disassembly mode that proceeds from the (+)-end.

Their inertness suggested the possibility of separating SubA-cleaved BiP oligomers from other cleavage products by size to enable their biochemical characterisation (*Figure 7D*). To this end, we incubated purified BiP protein at high concentration with SubA followed by preparative SEC to isolate the early-eluting oligomeric species. Subsequent analytical SEC revealed that the isolated, cleaved oligomers retained their expected elution profile (*Figure 7—figure supplement 2*) and

proved resistant to ATP-induced disassembly (stable over 3 hr at 26°C), and thus deemed suitable for further biochemical characterisation (*Figure 7—figure supplement 2*).

MABA-ADP (bound in the presence of $Mg^{2+}$) dissociated from stabilised (SubA-cleaved) BiP oligomers and untreated BiP at similar rates (*Figure 7E*). Addition of $Ca^{2+}$ ([$MgCl_2$]/[$CaCl_2$]=3) had a similar inhibitory effect on MABA-ADP dissociation from both samples. In contrast, under these conditions, Grp170 enhanced the rate of MABA-ADP release from stabilised oligomers only ~2.9-fold, whereas release from untreated BiP was enhanced ~12.3-fold (*Figure 7E*). The latter is consistent with the experiments shown above (*Figure 3A*). The relative weakness of Grp170 in encouraging nucleotide exchange on internal protomers of oligomeric BiP may also explain the apparent discrepancy between the marked effect of Grp170 on MABA-ADP exchange noted in the presence of $Ca^{2+}$ (*Figure 3A*) and the more modest effect of Grp170 on the dissociation of BiP-substrate interactions in the presence of $Ca^{2+}$ (*Figure 4C*), as assembly of BiP oligomers on the BLI probe would tend to limit the ability of Grp170 to accelerate complex dissociation to the single protomer attached on the (+)-end.

These results indicate (i) that passive nucleotide exchange occurs with normal rates on BiP oligomers but their conformationally trapped state renders them non-responsive to ATP binding, and (ii) that oligomers are largely resistant to Grp170-stimulated nucleotide release. Together with the finding that BiP oligomers disassemble from one end, these observations suggest a mechanism whereby the relative inertness of the internal protomers kinetically favours BiP oligomers over other BiP-substrate interactions in the $Ca^{2+}$-depleted ER.

## Discussion

The roles of BiP as an abundant effector chaperone and feedback regulator of the UPR render it an essential guardian of ER physiology. Both functions converge on conventional Hsp70 substrate-binding interactions and are subordinate to the kinetics of nucleotide turnover by BiP's ATPase activity. The finding that BiP's ATPase cycle is modulated by $Ca^{2+}$ fluctuations within the physiological range suggests a defined mechanism that links changes in $[Ca^{2+}]^{ER}$ to protein homeostasis. In the $Ca^{2+}$-depleted ER, BiP-substrate interactions are destabilised (*Figure 8*). The partitioning of substrates away from the only luminal Hsp70 favours trafficking of those substrates that would otherwise have been retained by BiP, whilst promoting de-repression of the UPR transducers, themselves BiP substrates. The perturbation to ER proteostasis is compounded by the polarity of BiP oligomers, which specifies their relative inertness to $Ca^{2+}$ fluctuations, thus kinetically driving BiP to oligomerise and depleting the pool of active monomers. Alongside the direct role played by $Ca^{2+}$ in the folding of certain secreted proteins, and the role of $Ca^{2+}$ binding to low-affinity sites in regulating the in vivo activity of parallel ER chaperones, the aforementioned mechanism goes some way to explain the well-established relationship between ER $Ca^{2+}$ depletion and the activation of ER stress signalling.

Comparison of cytosolic and ER Hsp70s suggests that regulation of BiP by $Ca^{2+}$ co-evolved with the functions of the mammalian ER in $Ca^{2+}$ storage and signalling. This is reflected in the relative insensitivity of cytosolic and bacterial Hsp70s to $Ca^{2+}$ and in apparent tuning of metazoan BiP to the range of physiological ER $Ca^{2+}$ fluctuations. In this vein, the yeast ER Hsp70 Kar2 is of special interest, as we found it to be much less sensitive to $Ca^{2+}$ than its metazoan counterparts, despite sharing ~72% sequence identity in the NBD. This correlates with the yeast vacuole, not the ER, serving as the major intracellular $Ca^{2+}$ store responsible for regulating $Ca^{2+}$ release (*Denis and Cyert, 2002*; *Halachmi and Eilam, 1989*; *Strayle et al., 1999*).

At high CaADP concentrations, the crystallised NBDs of all Hsp70's tested bind the cation-nucleotide complex indistinguishably. Solution-based assays suggest that subtle differences in protein dynamics, arising from the 30% sequence divergence between the NBD of metazoan BiP and its cytosolic counterparts, may account for their different sensitivity to $Ca^{2+}$. Regardless of its structural basis, it is intriguing to consider how this specialised feature of BiP (conserved from mammals to flies) improves fitness.

From the pancreatic acinus to killer T cells, secretion of proteins (stored in granules) is regulated by $Ca^{2+}$ release from the ER. The destabilising effect of ER $Ca^{2+}$ depletion on BiP-substrate complexes may have arisen as a response to situations in which the demands for bulk protein export from the ER (to replenish secreted protein stores) outweighs the benefits of stringent quality control (that would otherwise be imposed by more prolonged BiP-substrate interactions and ER retention).

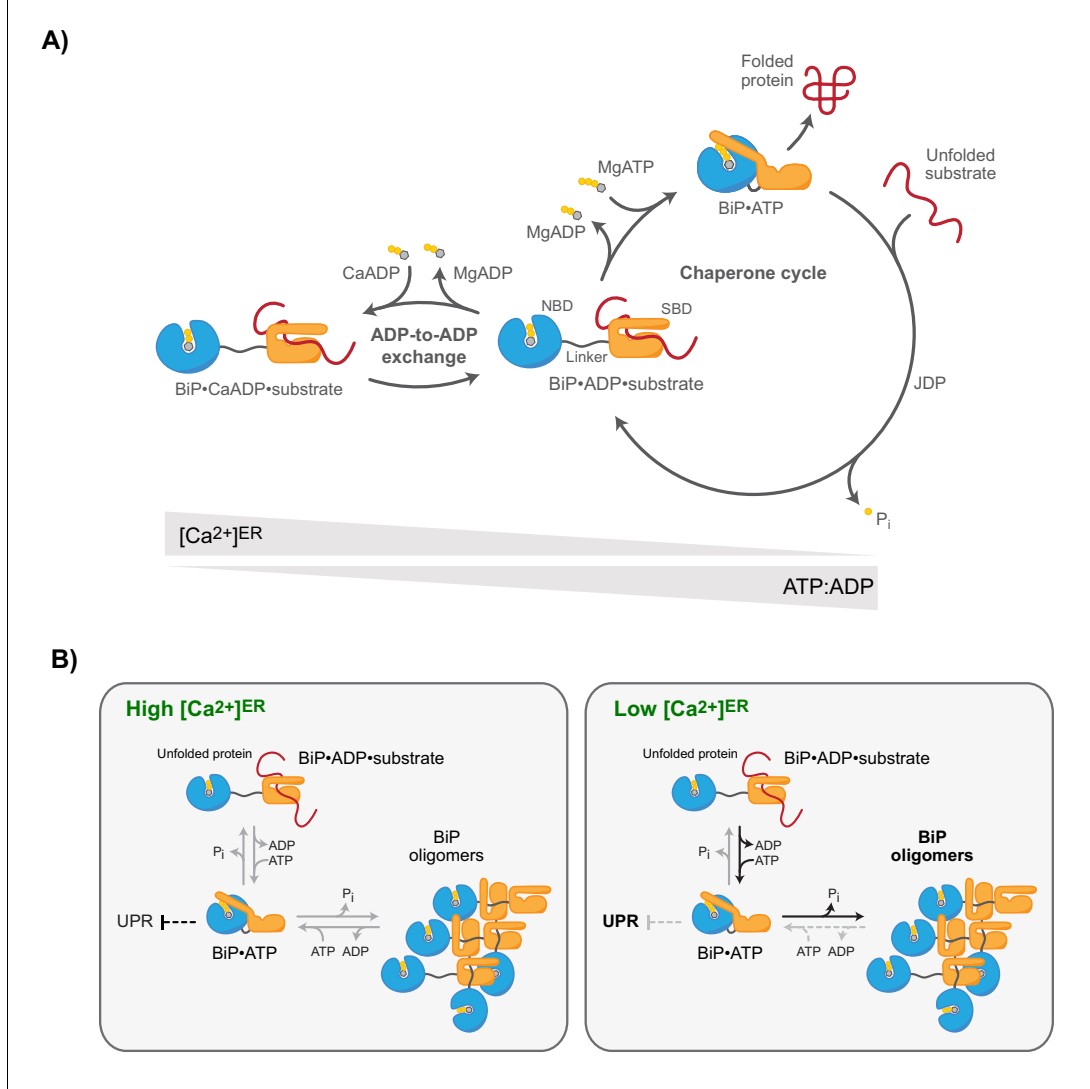

**Figure 8.** Ca$^{2+}$-dependent alternative fates of BiP during nucleotide exchange. (**A**) High [Ca$^{2+}$]$^{ER}$ stabilises BiP•ADP•substrate complexes. The interaction of BiP with its substrates is driven by an ATPase-dependent chaperone cycle, whereby ATP binding triggers substrate release and ATP hydrolysis (to ADP and orthophosphate; P$_i$) induces high-affinity substrate binding (right cycle). ADP competes with ATP for binding to BiP during post-hydrolysis nucleotide exchange, causing ADP-to-ADP exchange cycles that delay substrate release. Because Ca$^{2+}$ enhances the affinity of ADP for BiP, the substrate interaction-stabilising ADP-bound state of BiP is kinetically favoured by high [Ca$^{2+}$]$^{ER}$, whereas a decline in [Ca$^{2+}$]$^{ER}$ accelerates ADP-to-ATP exchange and substrate release. A decrease in the ATP:ADP ratio also promotes ADP-to-ADP exchange and extension of BiP-substrate interactions. (**B**) Observed redistribution of BiP upon ER Ca$^{2+}$ depletion. At high [Ca$^{2+}$]$^{ER}$ (resting state; left panel), the concentration of unfolded proteins governs the formation of BiP-substrate interactions and competing oligomerisation. BiP oligomers are conformationally locked (functionally inactive) assemblies formed by ATP hydrolysis-dependent substrate interactions amongst BiP proteins. BiP also participates (both indirectly and by direct interaction with UPR signal transducers) in repression of UPR signalling. The restricted ability of oligomeric BiP to respond allosterically to ATP binding (and the resistance of oligomeric BiP to NEF-stimulated ADP release; not shown) kinetically stabilises oligomers at the expense of heterodimeric BiP-substrate complexes in the Ca$^{2+}$-depleted ER (right panel). Faster substrate release from BiP at low [Ca$^{2+}$]$^{ER}$ and lower BiP availability (due to enhanced sequestration into oligomers) likely favours de-repression of UPR signalling.

The online version of this article includes the following figure supplement(s) for figure 8:

**Figure supplement 1.** A speculative model of the fitness-promoting features of the observed selective Ca$^{2+}$ sensitivity of metazoan ER-localised Hsp70s.

ER quality control responds continuously to the thermodynamic instability of its substrates (*Sekijima et al., 2005*). The dissociation of BiP from a class of not yet natively assembled but export-ready substrates during Ca$^{2+}$ signalling events may thus reflect a mechanism for tuning the stringency of quality control to physiological demands.

Activation of the UPR transducers IRE1, PERK, and ATF6 is a prominent consequence of ER Ca$^{2+}$ depletion. Disagreements on the details notwithstanding, most models for UPR regulation acknowledge the importance of BiP-substrate interactions (*Karagöz et al., 2019*). In one model, BiP is thought to bind directly to the regulatory luminal domain of the transducers via typical BiP-substrate interactions, to keep them in a repressed state (*Amin-Wetzel et al., 2019*; *Amin-Wetzel et al., 2017*). Competing substrates displace BiP from the signal transducers resulting in their activation. The findings presented here imply that declining ER Ca$^{2+}$ favours dissociation of BiP from UPR signal transducers, which is consistent with previous pulldown experiments showing release of BiP from PERK and ATF6 upon ER Ca$^{2+}$ depletion (*Bertolotti et al., 2000*; *Shen et al., 2002*). Importantly, BiP dissociation under these conditions is complete on a timescale of minutes and does not require an ongoing supply of nascent unfolded proteins, placing it upstream of any proteostatic challenge arising directly from Ca$^{2+}$ depletion. Similarly, dissociation of BiP from substrates may fuel the supply of activating ligands of the UPR transducers (*Karagöz et al., 2019*). Physiological fluctuations of Ca$^{2+}$ levels are often oscillatory (*Berridge et al., 2000*; *Dupont et al., 2011*; *Raffaello et al., 2016*; *Thomas et al., 1996*). The Ca$^{2+}$-entrained pulses of UPR signalling arising from the specialised sensitivity of BiP to Ca$^{2+}$ may have evolved to tune gene expression programs that build secretory capacity to the secretory activity of cells (*Figure 8—figure supplement 1*). Given the reversibility of the Ca$^{2+}$ effect on BiP and the transient nature of physiological Ca$^{2+}$ fluctuations, such conditioning of the ER is likely attained without a significant cost of protein aggregation and proteotoxicity.

Our findings indicate that Ca$^{2+}$ acts mainly on the nucleotide exchange phase of BiP's chaperone cycle. After post-hydrolysis ADP release, ATP and ADP compete for binding to BiP (*Figure 8*). Depending on the outcome of this competition BiP either enters a futile nucleotide exchange cycle through ADP re-binding, or acquires ATP, which triggers substrate release. Ca$^{2+}$ enhances the affinity of ADP for BiP and thereby promotes the ADP-bound state during nucleotide exchange. The efficiency of ADP-to-ATP exchange and rates of substrate release are thus inversely related to [Ca$^{2+}$]. As BiP requires MgATP to re-enter its chaperone cycle (*Kassenbrock and Kelly, 1989*), the rate at which ATP hydrolysis-dependent interactions with substrates are formed remain largely unaffected by Ca$^{2+}$. This rationalises the observed consequences of lowering free Ca$^{2+}$ levels in the ER on the interactions of BiP with its substrates: a decrease in [Ca$^{2+}$] enhances the effective rate of ATP-induced substrate release from BiP relative to the rate of substrate binding, leading to a net decline of BiP•substrate complexes.

Our structural data show that Ca$^{2+}$ binds in the nucleotide-binding cleft of the NBD and participates in coordinating the phosphate groups of ADP. This sufficiently explains the observed cooperative binding mode of Ca$^{2+}$ and nucleotide to BiP. Although none of our functional experiments indicated that Ca$^{2+}$ exerts an allosteric effect on BiP through a mechanism other than by replacing Mg$^{2+}$ during nucleotide binding, an additional Ca$^{2+}$ ion was present on the surface in all our Hsp70 NBD structures. Ca$^{2+}$ binding to this second site is likely an experimental artefact favoured by the high [Ca$^{2+}$] in the crystallisation solution and crystal packing. Moreover, mutation of the second metal-binding site did not affect the responsiveness of BiP to Ca$^{2+}$ in ADP release experiments, rendering its relevance to any allosteric regulation of BiP by Ca$^{2+}$ improbable.

Chaperones, in particular BiP, are the most abundant ATPases in the ER (*Bakunts et al., 2017*; *Dierks et al., 1996*; *Itzhak et al., 2016*). Increasing protein folding load could thus boost ATP consumption by chaperones and temporarily lower the luminal ATP:ADP ratio. Coupling regulation of BiP by Ca$^{2+}$ to the ATP:ADP ratio may therefore save energy at basally high luminal [Ca$^{2+}$] by amplifying the effect of ADP as a competitor for ATP and thus lowering BiP's ATPase activity. Similar principles may apply to chaperone systems in other cellular compartments as well (although there the impact of Ca$^{2+}$ is most likely negligible).

In contrast, ER Ca$^{2+}$ depletion might enhance BiP's ATPase activity. The cost of this to the ER energy balance may be compensated (at least transiently) by an increase in ATP import into the organelle (*Vishnu et al., 2014*; *Yong et al., 2019*). Furthermore, the effect on global energetics is mitigated by PERK activation and the attendant decrease in energetically expensive protein

synthesis. In this sense, it is interesting to speculate that BiP's responsiveness to $Ca^{2+}$ is an animal cell feature that co-evolved with the acquisition of PERK.

Likewise, BiP oligomerisation may have evolved to alleviate the consequences of enhanced ATP consumption at low $Ca^{2+}$ levels by sequestering BiP in inactive complexes with relatively slow turnover rates under these conditions. Mechanistically, this can be explained by the finding that BiP oligomer disassembly proceeds in a strictly directional manner and thus might be relatively slow compared to the one-step dissociation of heterodimeric BiP•substrate complexes. This difference is likely enhanced at low $[Ca^{2+}]$ when ADP-to-ATP exchange is more efficient and the unidirectional, step-wise dissociation of BiP oligomers imposes an even greater relative kinetic penalty, causing a redistribution of BiP from heterodimeric BiP•substrate complexes to oligomers. Furthermore, conformationally trapped oligomeric BiP is largely resistant to stimulated ADP release by Grp170, the most abundant NEF in the ER (*Behnke et al., 2015*; *Weitzmann et al., 2007*). Grp170 belongs to the Hsp110 class of NEFs, which bind Hsp70s via a large contact area (*Polier et al., 2008*). It is therefore likely that part of the Grp170-binding site on BiP is sterically occluded in oligomers. The differential action of Grp170 (and potentially other co-factors) on BiP•substrate complexes and oligomers likely contributes to the altered BiP dynamics at varying $Ca^{2+}$ levels.

Changes in BiP's substrate interaction kinetics are observed within minutes of ER $Ca^{2+}$ depletion (*Preissler et al., 2015*). The dissociation of substrates from BiP under conditions of low $[Ca^{2+}]^{ER}$ may allow some of them to escape retention in the ER and enhance their secretion. This is supported by the behaviour of the BiP-interacting TCRα reporters, which migrated to the cell surface after release of ER $Ca^{2+}$ (*Figure 1* and *Suzuki et al., 1991*). TCRα trafficking to the cell surface is not observed under conditions that primarily perturb protein folding homeostasis, suggesting that this effect of ER $Ca^{2+}$ depletion is not a consequence of downstream proteostatic challenge. However, the dominance of the mechanistic link proposed here for TCRα's itinerary remains unproven, as we are unable to establish a system in which BiP's essential role as a chaperone is uncoupled from its responsiveness to $Ca^{2+}$. Moreover, destabilisation of BiP-substrate interactions is likely not the only mechanistic connection between ER $Ca^{2+}$ release, enhanced protein trafficking, and UPR induction. The downstream perturbation to protein folding arising from substrate release and partitioning of BiP to oligomers may contribute, especially under experimental conditions of protracted ER $Ca^{2+}$ depletion following lengthy application of $Ca^{2+}$-mobilising agents or irreversible uptake inhibitors. The action of such parallel mechanism is evident in the escape of misfolded prion precursor protein from the ER quality control upon proteostatic challenge with a reducing agent (*Satpute-Krishnan et al., 2014*). It should be interesting to define in molecular terms the basis for other condition-dependent modulation of ER quality control.

## Materials and methods

**Key resources table**

| Reagent type (species) or resource | Designation | Source or reference | Identifiers | Additional information |
|---|---|---|---|---|
| Strain, strain background (*Escherichia coli*) | M15 | Qiagen | | Recombinant protein expression |
| Strain, strain background (*Escherichia coli*) | BL21 T7 Express lysY/Iq cells | New England Biolabs | Cat. #: C3013 | Recombinant protein expression |
| Cell line (*Cricetulus griseus*) | CHO-K1 | ATCC | RRID:CVCL_0214 | |
| Cell line (*Cricetulus griseus*) | CHO-K1 CHOP::GFP XBP1::Turquoise (clone S21) | PMID:27812215 | CHO-K1 S21 | CHO CHOP::GFP, XBP1s::Turquoise dual UPR reporter cell line; a derivative of RRID:CVCL_0214 |
| Cell line (*Homo sapiens*) | HEK293T | ATCC | RRID:CVCL_0063 | |

*Continued on next page*

*Continued*

| Reagent type (species) or resource | Designation | Source or reference | Identifiers | Additional information |
|---|---|---|---|---|
| Cell line (*Cricetulus griseus*) | CHO-K1 TCRα (clone 24) | This study | | Derivative of CHO-K1 cell line |
| Cell line (*Cricetulus griseus*) | CHO-K1 TCRα-N/Q (clone 12) | This study | | Derivative of CHO-K1 cell line |
| Antibody | Mouse monoclonal anti-FLAG M1; ANTI-FLAG M1 | Sigma | Cat. #: F3040 | Flow cytometry (1:500), WB (1:1000) |
| Antibody | Goat polyclonal anti-mouse IgG; Alexa Fluor 488 AffiniPure Goat Anti-Mouse IgG (H+L) | Jackson ImmunoResearch | Cat. #: 115-545-146 | Flow cytometry (1:750) |
| Antibody | Chicken polyclonal anti-BiP IgY | PMID:23589496 | | WB (1:2000) |
| Antibody | Donkey polyclonal anti-chicken IgY; IRDye 800CW Donkey anti-Chicken Secondary Antibody | Li-Cor | Cat. #: 926–32218 | WB (1:2000) |
| Antibody | Goat polyclonal anti-mouse IgG (H+L); IRDye 800CW Goat anti-Mouse IgG Secondary Antibody | Li-Cor | Cat. #: 926–32210 | WB (1:2000) |
| Peptide, recombinant protein | P15; ALLLSAPRRGAGKK | GenScript (custom synthesis) | | Biotinylated on the C-terminal lysine |
| Peptide, recombinant protein | NR; NRLLLTG | GenScript (custom synthesis) | | Carrying a fluorescein moiety at the N-terminus |
| Chemical compound, drug | Tunicamycin; Tm | Melford | Cat. #: T2250 | |
| Chemical compound, drug | Thapsigargin; Tg | Calbiochem | Cat. #: 586005 | |
| Chemical compound, drug | A23187 | Sigma | Cat. #: C7522 | |
| Chemical compound, drug | 2-Deoxy-D-glucose; 2DG | ACROS Organics | Cat. #: D6134 | |
| Chemical compound, drug | PERK inhibitor; PERKi | GSK | Cat. #: GSK2606414 | |
| Chemical compound, drug | Brefeldin A; BFA | LC Laboratories | Cat. #: B-8500 | |
| Chemical compound, drug | Puromycin | Cayman Chemical | Cat. #: 13884 | |
| Chemical compound, drug | TransIT-293 Transfection Reagent | Mirus | Cat. #: MIR 2704 | |
| Chemical compound, drug | Hexokinase Type F-300 | Sigma | Cat. #: H4502 | |
| Chemical compound, drug | MABA-ADP; 8-[(4-Amino) butyl]-amino-ADP-MANT | Jena Bioscience | Cat. #: NU-893-MNT | |
| Chemical compound, drug | MABA-ATP; 8-[(4-Amino) butyl]-amino-ATP-MANT | Jena Bioscience | Cat. #: NU-806-MNT | |
| Chemical compound, drug | SYPRO orange protein gel stain | Invitrogen | Cat. #: S6651 | |
| Software, algorithm | Prism | GraphPad | | |

*Continued on next page*

*Continued*

| Reagent type (species) or resource | Designation | Source or reference | Identifiers | Additional information |
|---|---|---|---|---|
| Software, algorithm | XIA2 | DOI: 10.1107/S0021889809045701 | | |
| Software, algorithm | Mosflm | PMID:21460445 | | |
| Software, algorithm | XDS | PMID:20124692 | | |
| Software, algorithm | Pointless | PMID:21460446 | | |
| Software, algorithm | Aimless | PMID:21460441 | | |
| Software, algorithm | Phaser | PMID:19461840 | | |
| Software, algorithm | COOT | PMID:20383002 | | |
| Software, algorithm | refmac5 | PMID:11134934 | | |
| Software, algorithm | phenix.refine | PMID:31588918 | | |
| Software, algorithm | UCSF Chimera | PMID:15264254 | | |
| Software, algorithm | PyMOL | PyMOL | | Version 1.3 educational |
| Software, algorithm | CheckMyMetal | PMID:28291757 | | |
| Software, algorithm | phenix.polder | PMID:28177311 | | |

## DNA plasmids

The plasmids used in this study have been described previously or were generated by standard molecular cloning techniques and are listed in *Supplementary file 7*.

## Mammalian cell lines

CHO-K1 cells (ATCC CCL-61) were phenotypically validated as proline auxotrophs and their Chinese hamster (*Cricetulus griseus*) origin was confirmed by genomic sequencing. The cells were grown on cell culture dishes or multi-well plates (Corning) at 37°C and 5% $CO_2$ in Nutrient mixture F-12 Ham (Sigma-Aldrich) supplemented with 10% (v/v) serum (FetalClone II; HyClone), 1 × penicillin–strepto-mycin (Sigma-Aldrich), and 2 mM L-glutamine (Sigma-Aldrich). Cell lines were randomly tested for mycoplasma contamination using the MycoAlert Mycoplasma Detection Kit (Lonza). All experiments were performed at cell densities of 60–90% confluence. Where indicated, cells were treated with 0.5 µM thapsigargin (Tg; Calbiochem), 2.5 µg/ml tunicamycin (Tm; Melford), 10 µM A23187 (Sigma, C7522), 3 mM 2-deoxy-D-glucose (2DG; ACROS Organics), 2 µM PERK inhibitor (PERKi; gift from GSK, GSK2606414), and 10 µg/ml brefeldin A (BFA; LC Laboratories, B-8500). All compounds were diluted freshly in pre-warmed medium and applied to the cells by medium exchange.

The UPR reporter cell line CHO-K1 *CHOP::GFP XBP1::Turquoise* (clone S21) has been described previously (*Sekine et al., 2016*). Cell lines stably expressing TCRα and TCRα-N/Q were generated by targeting CHO-K1 cells with puromycin-resistant retrovirus. For that, HEK293T cells (ATCC CRL-3216) were cultured in Dulbecco's modified Eagle's medium (Sigma) supplemented as described above. The cells were split onto 6-cm dishes 24 hr prior to co-transfection of pBABE Puro plasmids encoding TCRα or TCRα-N/Q (plasmids UK2469 and UK2520, respectively; *Supplementary file 7*) with VSV-G retroviral packaging vectors, using TransIT-293 Transfection Reagent (Mirus) according to the manufacturer's instructions. Sixteen hours after transfection, medium was changed to medium supplemented with 1% BSA (Sigma). Retroviral infections were

performed following a 24 hr incubation period by diluting 0.45 µm filter-sterilised cell culture supernatants at a 1:1 ratio with CHO-K1 cell medium supplemented with 10 µg/ml polybrene (8 ml final volume) and adding this preparation to target CHO-K1 cells ($1 \times 10^6$ cells seeded onto 10-cm dishes 24 hr prior to infection). After 8 hr, the viral supernatants were replaced with fresh medium. The cells were split 48 hr later into medium supplemented with 6 µg/ml puromycin (Calbiochem). After further 24 hr, the medium was changed to medium supplemented with 8 µg/ml puromycin. The medium was changed every third day until puromycin-resistant colonies were visible. Individual clones were isolated and tested for reporter expression. CHO-K1 TCRα (clone 24) and CHO-K1 TCRα-N/Q (clone 12) were used for all experiments shown.

## Flow cytometry

To analyse expression of TCRα and TCRα-N/Q reporters on the cell surface, stable CHO-K1 clones (see above) or parental cells were grown on six-well plates to 80–90% confluence and treated as indicated. The cells were then washed with 1 ml PBS and incubated in 0.5 ml PBS + 2 mM EDTA for 5 min at 37°C. Afterwards, the cells were transferred to a 1.2 ml tube (STARLAB, cat. # I1412-7400) and pelleted for 3 min in a swing-out rotor at 1000 rpm in a tabletop centrifuge (Allegra X-30R, Beckman Coulter). The supernatant was aspirated and cells were suspended in the remaining liquid to $1 \times 10^5 – 1 \times 10^7$ cells/100 µl. The cells were washed with 1 ml ice-cold TBS solution [TBS + 3% bovine serum albumin (BSA) + 2 mM $CaCl_2$ + 0.1% $NaN_3$] and pelleted as above. The supernatant was removed entirely and cells were suspended in 100 µl of the same TBS solution containing mouse monoclonal ANTI-FLAG M1 antibodies (Sigma, cat. # F3040) at a 1:500 dilution. The cells were incubated 30 min on ice while gently mixing every 10 min. After the incubation time, cells were washed twice in 1 ml TBS solution and suspended in TBS solution containing Alexa Fluor 488 Goat anti-mouse IgG (Jackson ImmunoResearch, cat. # 115-545-146; 0.75 mg/ml stock in 50% glycerol) at a 1:750 dilution for 30 min on ice in the dark as above. The cells were pelleted, washed with TBS solution and suspended in 400 µl TBS solution for flow cytometry analysis. For analysis of UPR reporter induction, CHO-K1 S21 cells were treated as indicated, washed with PBS and collected in PBS containing 4 mM ethylenediaminetetraacetic acid (EDTA).

Single-cell fluorescence signals (20,000/sample) were analysed by dual-channel flow cytometry with an LSRFortessa cell analyser (BD Biosciences). GFP fluorescence and Alexa Fluor 488 fluorescence was detected with excitation laser 488 nm and a 530/30 emission filter and Turquoise fluorescence was detected with excitation laser 405 nm and a 450/50 emission filter. Gating for live cells was based on FSC-A/SSC-A and for singlets was based on FSC-W/SSC-A. Data were processed using FlowJo and median reporter analysis was performed using Prism 6.0e (GraphPad). All events displayed were considered for analysis.

## Immonoprecipitation (IP) and immunoblotting

Mammalian cells were grown on 10-cm dishes and treated as indicated. The dishes were then placed on ice and washed with ice-cold PBS, and cells were detached in PBS containing 1 mM EDTA using a cell scraper. The cells were sedimented for 5 min at $370 \times g$ at 4°C and lysed in HG lysis buffer (20 mM HEPES-KOH pH 7.4, 150 mM NaCl, 2 mM $MgCl_2$, 10 mM D-glucose, 10% glycerol, 1% Triton X-100) containing protease inhibitors [2 mM phenylmethylsulphonyl fluoride (PMSF), 4 µg/ml pepstatin, 4 µg/ml leupeptin, 8 µg/ml aprotinin] with 100 U/ml hexokinase (from *Saccharomyces cerevisiae* type F-300; Sigma) for 10 min on ice. To release BiP from substrates, hexokinase was replaced by 1 mM ATP during lysis. The lysates were cleared for 10 min at $21,000 \times g$ at 4°C and Bio-Rad protein assay reagent (Bio-Rad) was used to determine the protein concentrations of lysates. Equal volumes of normalised lysates were incubated for 2 hr at 4°C with 20 µl UltraLink Hydrazine Resin (Pierce, 53149) on which hamster BiP-specific chicken IgY antibodies had been covalently immobilised (*Preissler et al., 2015*). The beads were recovered by centrifugation for 1 min at $1,000 \times g$ and washed three times (for 5 min) with HG lysis buffer. The proteins were eluted in $2 \times$ SDS sample buffer and denatured for 10 min at 70°C. Equal volumes of the samples were applied to SDS-PAGE and proteins were transferred onto PVDF membranes. The membranes were blocked with 5% (w/v) dried skimmed milk in TBS and incubated with primary antibodies followed by IRDye fluorescently labelled secondary antibodies (LiCor). The membranes were scanned with an Odyssey near-infrared imager (LiCor). Primary antibodies against hamster BiP [chicken anti-BiP IgY; (*Avezov et al., 2013*)]

and ANTI-FLAG M1 (see above) were used. The relevant signals were quantified with ImageJ (NIH) and analysed with Prism 8.4 (GraphPad).

## Protein purification

### BiP and stabilised oligomers

Wild-type (UK173) and mutant (UK838, UK2819, UK2820) Chinese hamster BiP proteins with an N-terminal His6-tag were expressed in M15 *E. coli* cells (Qiagen). The bacterial cultures were grown in LB medium with 100 µg/ml ampicillin and 50 µg/ml kanamycin at 37°C to an $OD_{600 \, nm}$ of 0.8 and expression was induced with 1 mM isopropylthio β-D-1-galactopyranoside (IPTG). The cells were further grown for 6 hr at 37°C, collected and lysed with a high-pressure homogeniser (EmulsiFlex-C3; Avestin) in buffer A (50 mM Tris–HCl pH 8, 500 mM NaCl, 1 mM $MgCl_2$, 10% glycerol, 20 mM imidazole) containing protease inhibitors (1 mM PMSF, 2 µg/ml pepstatin, 8 µg/ml aprotinin, 2 µM leupeptin) and 0.1 mg/ml DNaseI. The lysates were cleared for 30 min at 45,000 × *g* at 4°C and incubated with 1 ml of Ni-NTA agarose beads (Qiagen) per 1 l of expression culture for 2 hr rotating at 4°C. Afterwards, the beads was transferred to a gravity-flow Econo column (49 ml volume; Bio-Rad) and washed with two column volumes (CV) buffer B (50 mM Tris–HCl pH 8.0, 500 mM NaCl, 10% glycerol, 30 mM imidazole), 2 CV buffer C (50 mM Tris–HCl pH 8.0, 300 mM NaCl, 10 mM imidazole, 5 mM β-mercaptoethanol) and buffer C sequentially supplemented with (i) 1 M NaCl, (ii) 10 mM $MgCl_2$ + 3 mM ATP, (iii) 0.5 M Tris–HCl pH 7.4, or (iv) 35 mM imidazole (2 CV each). The BiP proteins were then eluted with buffer D (50 mM Tris–HCl pH 8.0, 300 mM NaCl, 5 mM β-mercaptoethanol, 250 mM imidazole), dialysed against HKM (50 mM HEPES-KOH pH 7.4, 150 mM KCl, 10 mM $MgCl_2$) and concentrated with 30-kDa MWCO centrifugal filters (Amicon Ultra, Merck Millipore). The proteins were flash-frozen in aliquots and stored at −80°C. To obtain nucleotide-free preparations, the purified proteins were additionally dialysed 24 hr at 4°C against 50 mM HEPES-KOH pH 7.4, 150 mM KCl, 4 mM EDTA and twice against 50 mM HEPES-KOH pH 7.4, 150 mM KCl before concentration and freezing.

Stabilised BiP oligomers were generated from purified BiP protein (UK173). The proteins were concentrated in HKM buffer to ~16 mg/ml and SubA protease (produced in-house) (*Paton et al., 2006*) was added at 50 ng/µl. After incubation for 40 min at 26°C to allow for complete cleavage, the sample was cooled on ice for 5 min. The solution was then cleared through a 0.22 µm pore size centrifuge tube filter (Spin-X, Corning) and injected onto a Superdex 200 Increase 10/300 GL column (GE Healthcare) equilibrated in 50 mM HEPES-KOH pH 7.4, 150 mM KCl. The early eluting peak fractions containing oligomeric BiP species were immediately pooled, concentrated and frozen in aliquots.

### Grp170 and authentic Hsp70s

Full-length human Grp170 (UK2225), BiP from *Drosophila melanogaster* (UK2354) as well as human Hsp70 (UK2510) and Hsc70 (UK2511) were expressed as fusion proteins with an N-terminal His6-Smt3 in *E. coli* BL21 T7 Express lysY/Iq cells (New England BioLabs, cat. # C3013). The cells were grown in LB medium containing 50 µg/ml kanamycin at 37°C until $OD_{600 \, nm}$ 0.5. The cells were then transferred to 20°C and expression was induced after 20 min with 0.5 mM IPTG. After incubation for 14 hr, the cells were collected by centrifugation and suspended in lysis buffer E (50 mM Tris-HCl pH 8, 500 mM NaCl, 20 mM imidazole, 10% glycerol) containing protease inhibitors (1 mM PMSF, 2 µg/ml pepstatin, 8 µg/ml aprotinin, 2 µM leupeptin). Before lysis 1 mM $MgCl_2$, 0.1 mM tris (2-carboxyethyl)phosphine (TCEP), and 0.1 mg/ml DNaseI were added and lysis was performed with an EmulsiFlex-C3 high-pressure homogeniser. The lysates were cleared for 30 min at 45,000 × *g* at 4°C and incubated with 0.6 ml of Ni-NTA agarose beads (Qiagen) per 1 l of expression culture (1 ml beads per 1 l expression culture was used in case of *Drosophila* BiP, Hsp70 and Hsc70) for 1 hr rotating at 4°C. Afterwards, the beads were transferred to a 49 ml gravity-flow column and washed with 2 CV lysis buffer E, 2 CV buffer F (25 mM Tris–HCl pH 8, 150 mM NaCl, 10 mM imidazole, 5 mM β-mercaptoethanol), and buffer F sequentially supplemented with (i) 1 M NaCl, (ii) 10 mM $MgCl_2$ + 3 mM ATP + 0.01 mg/ml RNaseA, or (iii) 0.5 M Tris–HCl pH 8 + 1 mM TCEP (2 CV each), followed by 2 CV buffer F. Proteins were eluted by on-column cleavage with 1.5 µg/ml Ulp1 protease carrying a C-terminal StrepII-tag in 2.5 bed volumes buffer TNT-Iz15 (25 mM Tris–HCl pH 8, 150 mM NaCl, 1 mM TCEP, 15 mM imidazole) overnight at 4°C. The eluate was collected, retained cleavage products

were washed off the beads with TNT-Iz15, and all fractions were pooled. The combined eluate was diluted 1:3 in 25 mM Tris-HCl pH 8 + 0.1 mM TCEP, passed through a 0.22 µm filter, and further purified by anion exchange chromatography using a 6-ml RESOURCE Q column (GE Healthcare) equilibrated in 95% AEX-A (25 mM Tris–HCl pH 8, 25 mM NaCl, 0.1 mM TCEP) and 5% AEX-B (25 mM Tris–HCl pH 8, 1 M NaCl, 0.1 mM TCEP). Proteins were eluted by applying a gradient from 5% to 50% AEX-B in 25 CV at 6 ml/min. The elution peak fractions corresponding to Grp170 were pooled and 1 mM TCEP was added. After buffer exchange to HKT (50 mM HEPES-KOH pH 7.4, 150 mM KCl, 1 mM TCEP) and concentration using 30-kDa MWCO centrifugal filters, the proteins were frozen in aliquots and stored at −80°C. BiP from *Drosophila melanogaster* (UK2354) and human Hsp70 (UK2510; natural variant E110D) and Hsc70 (UK2511) were all expressed as fusion proteins with an N-terminal His6-Smt3 and purified likewise.

DnaK (from *Escherichia coli*) with an N-terminal His6-Smt3 (UK2243) was expressed and purified likewise, except that after anion exchange chromatography the pooled eluate was adjusted to 500 mM NaCl and 40 mM imidazole. The eluate was then incubated with 1 ml Ni-NTA agarose for 1 hr rotating at 4°C (to remove traces of uncleaved protein). The beads were then collected by centrifugation at 100 × *g* for 2 min and the supernatant was transferred to a 30-kDa MWCO centrifugal filter for buffer exchange to HKT and protein concentration.

## NBDs of BiP, Hsp70, and Hsc70 for crystallisation

All NBDs were expressed as fusion proteins with a N-terminal His6-Smt3. The NBD of hamster BiP (UK2039) was produced in M15 *E. coli* cells grown in LB medium containing 50 µg/ml kanamycin and 100 mg/ml ampicillin. The NBDs of human Hsp70 (UK2532; natural variant E110D) and human Hsc70 (UK2533) were produced in *E. coli* BL21 T7 Express lysY/Iq cells grown in LB medium containing 50 µg/ml kanamycin. The cells were grown at 37°C to $OD_{600\,nm}$ 0.5, transferred to 22°C for 20 min, and expression was induced with 0.5 mM IPTG. After incubation for 14 hr at 22°C, the cells were collected by centrifugation and suspended in lysis buffer G (50 mM Tris-HCl pH 8, 500 mM NaCl, 20 mM imidazole) containing protease inhibitors (1 mM PMSF, 2 µg/ml pepstatin, 8 µg/ml aprotinin, 2 µM leupeptin), 1 mM $MgCl_2$, 0.1 mM TCEP, and 0.1 mg/ml DNaseI. The cells were lysed with an EmulsiFlex-C3 high-pressure homogeniser and lysates were centrifuged for 30 min at 45,000 × *g* at 4°C. The cleared lysates were incubated with 1 ml Ni-NTA agarose beads per 1 l expression culture for 30 min rotating at 4°C. The beads were then transferred to a 49-ml gravity-flow column, washed in 2 CV lysis buffer G, 2 CV buffer H (50 mM Tris-HCl pH 8, 500 mM NaCl, 30 mM imidazole, 5 mM β-mercaptoethanol), 2 CV buffer I (50 mM Tris-HCl pH 8, 300 mM NaCl, 10 mM imidazole, 5 mM β-mercaptoethanol), and with buffer I supplemented sequentially with (i) 1 M NaCl, (ii) 10 mM $MgCl_2$ + 3 mM ATP + 0.01 mg/ml RNaseA, or (iii) 0.5 M Tris–HCl pH 8 + 1 mM TCEP (2 CV each), followed by 2 CV buffer TNT-Iz10 (25 mM Tris–HCl pH 8, 150 mM NaCl, 1 mM TCEP, 10 mM imidazole). Proteins were eluted by on-column cleavage with 3.7 µg/ml Ulp1 protease carrying a C-terminal StrepII-tag in one bed volume buffer TNT-Iz10 overnight at 4°C. The eluate was collected, retained cleavage products were washed off the beads with buffer TNT-Iz10, and all fractions were pooled. To remove bound nucleotide, the proteins were dialysed for 6 hr against buffer J (25 mM Tris-HCl pH 8, 150 mM NaCl, 5 mM EDTA, 5 mM β-mercaptoethanol), 16 hr against fresh buffer J, 6 hr against buffer K (25 mM Tris-HCl pH 8, 100 mM NaCl, 5 mM β-mercaptoethanol), and 16 hr against fresh buffer K. The protein solutions were then diluted 1:2 with 25 mM Tris-Hcl pH 8 + 0.2 mM TCEP, filtered (0.22 µm filter), and applied to anion exchange chromatography using a 6-ml RESOURCE Q column equilibrated in 100% AEX-A1 (25 mM Tris–HCl pH 8, 75 mM NaCl, 0.2 mM TCEP). The flow-through fractions were collected and bound proteins were eluted by applying a gradient from 0% to 50% AEX-B1 (25 mM Tris–HCl pH 8, 1 M NaCl, 0.2 mM TCEP) in 25 CV at 5 ml/min. The fractions corresponding to the NBD proteins were pooled and further purified. In case of the NBDs of Hsp70 (UK2532) and Hsc70 (UK2533) the unbound material (flow-through of the loading step) was used for further purification. The proteins were concentrated, applied to gel-filtration using a Superdex 75 prep grade HiLoad 16/60 column (GE Healthcare) equilibrated in buffer K (10 mM Tris-HCl pH 8, 150 mM KCl, 0.2 mM TCEP), and proteins were eluted at 0.75 ml/min. The elution peak fractions were pooled, supplemented with 1 mM TCEP, and proteins were concentrated in 10-kDa MWCO centrifugal filter to ~1–1.5 mM. The protein preparations were frozen in aliquots and stored at −80°C for crystallisation and functional experiments.

## NBDs of *Drosophila* BiP, DnaK, and Kar2

The NBDs were expressed as fusion proteins with a N-terminal His6-Smt3. The NBD of *Drosophila* BiP (UK2534) was produced in M15 *E. coli* cells grown in LB medium containing 50 µg/ml kanamycin and 100 mg/ml ampicillin. The NBDs of DnaK (UK2531) and yeast Kar2 (UK2606) were produced in *E. coli* BL21 T7 Express lysY/Iq cells grown in LB medium containing 50 µg/ml kanamycin. The cells were grown at 37°C to $OD_{600\ nm}$ 0.5, transferred to 22°C for 20 min, and expression was induced with 0.5 mM IPTG. After incubation for 14 hr at 22°C, the cells were collected by centrifugation and suspended in lysis buffer E (50 mM Tris-HCl pH 8, 500 mM NaCl, 20 mM imidazole, 10% glycerol) containing protease inhibitors (1 mM PMSF, 2 µg/ml pepstatin, 8 µg/ml aprotinin, 2 µM leupeptin), 1 mM $MgCl_2$, 0.1 mM TCEP, and 0.1 mg/ml DNaseI. The cells were lysed with an EmulsiFlex-C3 high-pressure homogeniser and lysates were centrifuged for 30 min at 45,000 × *g* at 4°C. The cleared lysates were incubated with 1 ml Ni-NTA agarose beads per 1 l expression culture for 30 min rotating at 4°C. The beads were then transferred to a 49-ml gravity-flow column, washed in 2 CV lysis buffer E, 2 CV buffer H (50 mM Tris-HCl pH 8, 500 mM NaCl, 30 mM imidazole, 5 mM β-mercaptoethanol), 2 CV buffer I (50 mM Tris-HCl pH 8, 300 mM NaCl, 10 mM imidazole, 5 mM β-mercaptoethanol) and with buffer I supplemented sequentially with (i) 1 M NaCl, (ii) 10 mM $MgCl_2$ + 3 mM ATP + 0.01 mg/ml RNaseA, or (iii) 0.5 M Tris–HCl pH 8 + 1 mM TCEP (2 CV each), followed by 2 CV buffer TNT-Iz10 (25 mM Tris–HCl pH 8, 150 mM NaCl, 1 mM TCEP, 10 mM imidazole). Proteins were eluted by on-column cleavage with 3.7 µg/ml Ulp1 protease carrying a C-terminal StrepII-tag in one bed volume buffer TNT-Iz10 overnight at 4°C. The eluate was collected, retained cleavage products were washed off the beads with buffer TNT-Iz10, and all fractions were pooled. After buffer exchange to HKT (50 mM HEPES-KOH pH 7.4, 150 mM KCl, 1 mM TCEP) and concentration using 10 kDa MWCO centrifugal filters, the proteins were frozen in aliquots and stored at −80°C for functional experiments.

## BiP for oligomer crystallisation

Chinese hamster BiP$^{T229A-V461F}$ (residues 28–549) was expressed as an N-terminal His6-Smt3-tagged protein (UK2090). The purification protocol was the same as used for His6-Smt3-FICD purification described in *Perera et al., 2019* with minor modifications. The protein was expressed in M15 *E. coli* cells grown in LB medium supplemented with 100 µg/ml ampicillin and 50 µg/ml kanamycin. After anion exchange chromatography the protein was exchanged into 10 mM HEPES-KOH pH 7.4, 150 mM KCl, 2 mM $MgCl_2$, and 1 mM TCEP by gel filtration and concentrated to 21 mg/ml.

For crystallisation, BiP and an equimolar amount of human FICD$^{L258D-H363A}$ [residues 104–445; UK2093], purified as in *Perera et al., 2019* were diluted to 150 µM with TNKM (10 mM Tris-HCl pH 8.0, 100 mM NaCl, 50 mM KCl and 2 mM $MgCl_2$) and supplemented with 175 µM ATP. Note: FICD was excluded from the resulting crystals of oligomeric BiP (see below).

## Biotinylated J-domains

The J-domain of *E. coli* DnaJ (amino acids 1–72; UK2470) as well as the wild-type (UK1965; J) and H422Q mutant (UK1966; J$^{QPD}$) J-domain of mammalian ERdj6 (amino acids 390–455) were expressed with an AviTag-His6 sequence at the C-terminus in BL21 T7 Express lysY/Iq cells grown in LB medium containing 50 µg/ml kanamycin. The cells were grown at 37°C to $OD_{600\ nm}$ 0.5, transferred to 18°C for 20 min, and expression was induced with 0.5 mM IPTG. The cells were incubated for 16 hr at 18°C during which the recombinant proteins were biotinylated on the AviTag to sufficient degree by the endogenous *E. coli* BirA biotin ligase. Afterwards, the cells were collected by centrifugation and lysed in buffer G (50 mM Tris-HCl pH 8, 500 mM NaCl, 20 mM imidazole) containing protease inhibitors (1 mM PMSF, 2 µg/ml pepstatin, 8 µg/ml aprotinin, 2 µM leupeptin), 1 mM $MgCl_2$, 0.1 mM TCEP, 0.1 mg/ml DNaseI, and 0.02 mg/ml RNaseA using an EmulsiFlex-C3 high-pressure homogeniser. The lysates were cleared for 30 min at 45,000 × *g* at 4°C and the supernatants were incubated with 1 ml Ni-NTA agarose beads per 1 l expression culture for 2 hr rotating at 4°C. The beads were then transferred to a 49 ml gravity-flow column, washed with 1 CV buffer I (50 mM Tris-HCl pH 8, 300 mM NaCl, 10 mM imidazole, 5 mM β-mercaptoethanol) and with buffer I supplemented sequentially with (i) 1 M NaCl, (ii) 10 mM $MgCl_2$ + 3 mM ATP, or (iii) 0.5 M Tris–HCl pH 8 + 1 mM TCEP (2 CV each), followed by 2 CV buffer I containing 35 mM imidazole. Proteins were eluted in buffer J (50 mM Tris-HCl pH 8, 300 mM NaCl, 250 mM imidazole) and dialysed against TNM (50

mM Tris-HCl pH 8, 150 mM NaCl, 2 mM MgCl$_2$) for 16 hr at 4°C and for further 6 hr against fresh TNM. The protein solutions were frozen in aliquots and stored at −80°C for Bio-Layer interferometry experiments.

## X-ray crystallography

Concentrated NBD protein preparations (see above) of hamster BiP (UK2039), human Hsp70 (UK2532), and human Hsc70 (UK2533) were adjusted to ~50 mg/ml. The preparations were supplemented with an additional 7 mM CaCl$_2$ and 2 mM ADP. Crystallisation of NBDs was performed in 96-well sitting drop plates by combining 200 nl protein solution with 200 nl reservoir solution and equilibration at 20°C against 70 µl reservoir solution. Crystallisation of lid-truncated BiP oligomers was achieved by mixing 100 nl of protein solution and 100 nl reservoir solution. Diffraction quality crystals grew in the following solutions: BiP(NBD) — 26% PEG6000, 0.2 M CaCl$_2$, 0.1 M NaOAc pH 5; Hsp70(NBD) — 10% PEG8000, 0.1 M Tris pH 8.4, 0.2 M CaCl$_2$; Hsc70(NBD) — 24% PEG3350, 0.2 M CaOAc; apo BiP oligomer — 20% PEG3350, 0.1 M Bis-Tris Propane pH 7.5, 0.2 M Na$_3$Citrate; ADP-bound BiP oligomer — 20% PEG3350, 0.2 M K$_3$Citrate. Crystals were soaked in cryosolution (the well solutions containing additional 25% [v/v] glycerol) and snap frozen in liquid nitrogen. Diffraction data were collected at the Diamond Light Source beamlines i03, i04, and i24 (Diamond Light Source, United Kingdom; see *Table 1*) and processed by the XIA2 pipeline (*Winter, 2010*) implementing Mosflm (*Battye et al., 2011*) or XDS (*Kabsch, 2010*) for indexing and integration, Pointless (*Evans, 2011*) for space group determination, and Aimless (*Winn et al., 2011*) for scaling and merging. All structures were solved by molecular replacement in Phaser (*McCoy et al., 2007*). NBD datasets were solved by using BiP nucleotide binding domain (PDB 3IUC) as the search model. BiP oligomer structures were solved by searching for an ensemble of apo BiP's NBD and SBD, taken from PDB 6HAB. Manual model building was carried out in COOT (*Emsley et al., 2010*) and further refinements were performed in refmac5 (*Winn et al., 2001*) and phenix.refine (*Liebschner et al., 2019*). The final refinement statistics are summarised in *Table 1*. The structural graphs were generated with UCSF Chimera (*Pettersen et al., 2004*) and Pymol (version 1.3 educational). CheckMyMetal (*Zheng et al., 2017*) has been used for metal validation. The polder maps in *Figure 6—figure supplement 1A* were calculated using phenix.polder (*Liebschner et al., 2017*) by omitting ADP, Ca$^{2+}$, and Ca$^{2+}$-coordinated water molecules.

## NADH ATPase assay

BiP (UK173), J (UK1965), and J$^{QPD}$ (UK1966) proteins were diluted from concentrated stocks and exposed to ATP and β-NADH to start the reactions. Proteins were used at final concentrations as indicated. The reactions were performed in HK solution (50 mM HEPES-KOH pH 7.4, 100 mM KCl) containing 10 mM MgCl$_2$, 0.05% Triton X-100, 1 mM TCEP. The reactions also contained 4 mM ATP, 3 mM Phospho(enol)pyruvic acid (PEP; Sigma, cat. # P0564), 0.25 mM β-NADH (Sigma, cat. # N8129), and pyruvate kinase/lactic dehydrogenase enzyme mix (diluted final 1:45; Sigma, cat. # P0294). Where indicated 3 mM CaCl$_2$ was present. The reactions were carried out in a final volume of 110 µl in 384-well microplates (µClear; Grainer bio-one, cat. # 781096) and β-NADH absorbance at 340 nm (A$_{340}$) was measured over time at 25°C with a Clariostar plate reader (BMG Labtech). Linear regression analysis was performed with Prism 8 (GraphPad) and the ATP hydrolysis activity was calculated using the molar extinction coefficient of β-NADH (ε = 6220 M$^{-1}$ cm$^{-1}$).

## Malachite green ATPase assay

Malachite green (MG) reaction solution was prepared freshly by combining stock solutions of MG dye (0.111% malachite green, 6 N sulphuric acid), 7.5% ammonium molybdate, and 11% Tween 20 in a 5:1.25:0.1 ratio. Samples of BiP protein (UK173) at 5 µM were prepared in HK solution containing 3 mM MgCl$_2$ and 3 mM ATP in a final volume of 20 µl. Where indicated ADP and CaCl$_2$ was added. The samples were incubated for 1.5 hr at 26°C. Afterwards, 15 µl of each sample were diluted with 135 µl water on a 96-well plate, mixed with 50 µl MG reaction solution, and incubated for 2 min at room temperature. Twenty µl of a 34% sodium citrate solution were added to each well. After further 30 min incubation at room temperature, the absorbance at 623 nm (A$_{623}$) was measured with a plate reader (SpectraMax, Molecular Devices). Standard curves from serial dilutions of KH$_2$PO$_4$ served as a reference to calculate the specific ATPase activity.

## Nucleotide- binding assays

Nucleotide release and binding measurements were performed using the fluorescent nucleotide analogues MABA-ADP (Jena Bioscience, cat. # NU-893-MNT) and MABA-ATP (Jena Bioscience, cat. # NU-806-MNT) carrying a MANT moiety whose fluorescence increases upon binding to Hsp70s (*Theyssen et al., 1996*). The measurements were started by manually adding 55 µl of solution A to 5 µl of solution B (specified below) in a quartz cuvette (Hellma Analytics) and fluorescence (excitation 360 nm, emission 420 nm) was detected over time with a fluorescence spectrometer (Perkin Elmer LS55) at 26°C. The dead time between sample mixing and signal recording was ~ 1 s. All solutions contained 50 mM HEPES-KOH pH 7.4, 100 mM KCl.

To measure nucleotide release, 1.42 µM fluorescent nucleotide analogues were incubated with 1.42 µM proteins in solution A for at least 1 hr at 26°C to allow for complex formation (binding phase). Solution B contained assay buffer with 18 mM MgATP. Where specified, 15.6 µM Grp170 were added to solution B. The final concentrations after mixing (release phase) were 1.3 µM for each protein, 1.3 µM fluorescent nucleotides, and 1.5 mM MgATP. The solutions contained $MgCl_2$ and $CaCl_2$ where indicated (final concentrations are stated on the figures and in figure legends). For $Ca^{2+}$ titration experiments, $CaCl_2$ was present in solution A and final $[CaCl_2]$ are plotted on the graphs and were used to calculate $IC_{50}$ values. The dissociation rate constants ($k_{off}$) were determined by fitting the data to a one phase exponential function using Prism 8.4 (GraphPad). In experiments involving cytoplasmic Hsp70s, all solutions contained 0.2 mM TCEP (*Figure 5A*, *Figure 6C*, *Figure 6—figure supplement 2A*).

The association rate constants (*Figure 3C*) were determined by combining nucleotide-free BiP (UK173) or $BiP^{T229A}$ (UK838) proteins (in solution A) with MABA-ADP or MABA-ATP (in solution B) at the start of the measurement. Solutions A and B contained either 1 mM $MgCl_2$ or $CaCl_2$. Each association rate was calculated from six individual measurements, covering a final protein concentration range from 1.3 µM to 7.8 µM. For that, the data were fitted to a one phase exponential function and the observed rates ($k_{obs}$) were plotted against the protein concentration to calculate association rate constants ($k_{on}$) with the equation $k_{obs} = k_{on}[protein] + k_{off}$ using the dissociation rate constants ($k_{off}$) determined separately under the same conditions.

## Bio-Layer interferometry (BLI)

BLI experiments were performed on the FortéBio Octet RED96 System (Pall FortéBio) in buffer solution containing 50 mM HEPES-KOH pH 7.4, 100 mM KCl, 0.2 mM TCEP, and 0.05% Triton X-100 (HKTTx). Nucleotide and divalent cations were added as indicated. Streptavidin (SA)-coated biosensors (Pall FortéBio) were hydrated in HKTTx solution for at least 10 min prior to use. Experiments were conducted at 28°C and 700 rpm shake speed and binding signals were recorded at an acquisition rate of 10 Hz. BLI reactions were prepared in 200 µl volumes in 96-well microplates (greiner bio-one, cat. # 655209). The following assay steps were performed: After an initial equilibration step in HKTTx solution (I), JDPs (J, UK1965 or $J^{QPD}$, UK1966) carrying a biotinylated Avi-tag (each at 5 µM in HKTTx) were loaded until a binding signal of ~0.4 nm was reached. After a brief wash step in assay solution (III) the binding sites on the sensors were saturated with P15 peptide (ALLLSAPRRGAGKK, biotinylated on the C-terminal lysine; custom synthesised by GenScript, Piscataway, NJ) solubilised at 100 nM in HKTTx solution (IV). The sensors were then washed in HKTTx until a stable baseline signal was established (V). Association of BiP or other Hsp70s was performed in HKTTx solution containing 3 mM $MgCl_2$, 1.5 mM ATP and the indicated proteins at 5 µM (VI). To allow for dissociation, the sensors were transferred to protein-free assay solution with variable components as indicated (VII). The data were processed in Prism 8.4 (GraphPad) and dissociation rate constants were obtained by fitting to a one phase exponential function. When DnaK was analysed (*Figure 5C*), a JDP containing the J-domain of DnaJ was immobilised (UK2470).

## Differential scanning fluorimetry (DSF)

DSF measurements were performed on a Bio-Rad CFX96 Touch Real-Time PCR Detection System in 96-well plates (Bio-Rad Hard-Shell, cat. # HSP9601) sealed with transparent adhesive film (Bio-Rad Microseal 'B' Adhesive Sealer, cat. # MSB1001). Samples were prepared in a volume of 25 µl. The solutions contained 50 mM HEPES–KOH pH 7.4, 100 mM KCl, 1 mM TCEP and SYPRO orange protein gel stain at a 1:500 dilution of the manufacturer's stock (Invitrogen, cat. # S6651). The samples

also contained 5 µM NBD protein, 4 mM ADP, and 6 mM $MgCl_2$ or $CaCl_2$ as indicated. The plates were centrifuged for 1 min at $150 \times g$ before the experiment. Temperature scanning was performed in 0.5°C intervals from 20°C to 95°C and SYPRO orange fluorescence was detected. The data were then analysed in Prism 8.4 (GraphPad) and the melting temperature ($T_m$) was determined as the temperature corresponding to the global minimum of the negative first derivatives of the melt curves.

## Size-exclusion chromatography (SEC)

Samples for analytical SEC were prepared in SEC buffer (50 mM HEPES-KOH pH 7.4, 150 mM KCl, 10 mM $MgCl_2$) in a final volume of 25 µl. The samples were incubated for at least 20 min at 25°C before 10 µl were injected onto a SEC-3 HPLC column (300 Å pore size; Agilent Technologies, Santa Clara, CA) equilibrated with SEC buffer. The runs were performed at a flow rate of 0.3 ml/min at 26°C and peptide bond absorbance at 230 nm ($A_{230}$) was detected and plotted against the elution time. A gel filtration standard (Bio-Rad, cat. # 151–1901) was applied as a size reference and the elution peaks of Thyroglobulin (670 kDa), γ-globulin (158 kDa), Ovalbumin (44 kDa), and Myoglobulin (17 kDa) are indicated.

In the experiment shown in *Figure 7C*, the samples contained 40 µM BiP protein (UK173) and 1 mM ADP and the samples were incubated for 20 min at 24°C before SEC analysis. Where indicated 36 ng/µl SubA protease was added during the incubation period.

For the experiment shown in *Figure 7D*, BiP at 50 µM was incubated for 6 hr at 24°C with 1 mM ADP and 1.3 µM fluorescently labelled NR peptide (NRLLLTG carrying a fluorescein moiety at the N-terminus, custom synthesised by GenScript at >95% purity; *Yang et al., 2017*). Afterwards, the samples were incubated without or with SubA protease (36 ng/µl) for 30 min before SEC analysis. Where indicated 5 mM ATP was added 10 min before injection into the column. The final concentration of BiP in the samples was 45 µM. The fluorescein fluorescence signal (excitation 495 nm; emission 520 nm) of the peptide was recorded separately during the SEC runs.

For the experiment shown in *Figure 7—figure supplement 2*, untreated BiP protein or SubA-treated, purified BiP oligomers (both at 20 µM) were incubated without or with 4 mM ATP in SEC buffer solution at 26°C as indicated. The BiP oligomers had been thawed from frozen stocks before the experiment.

## t-tests

Unless stated otherwise, unpaired, parametric, *t*-tests were applied where indicated using GraphPad Prism 8.

# Acknowledgements

We thank the CIMR flow cytometry facility for technical support, the James Huntington group (CIMR) for access to BLI equipment, Matthias P Mayer (University of Heidelberg) for Hsp70 and Hsc70 expression plasmids, and Claes Andréasson (Stockholm University) for Grp170 expression plasmids. We are grateful to the Diamond Light Source for beamtime (proposals mx21426 and mx27082) and the staff of beamlines I03, I04 and I24 for assistance with data collection. We also thank Luka Smalinskaite for help with ATPase experiments as well as Alisa F Zyryanova and Lisa Neidhardt for comments on the manuscript.

# Additional information

#### Competing interests

David Ron: Reviewing editor, *eLife*. The other authors declare that no competing interests exist.

#### Funding

| Funder | Grant reference number | Author |
| --- | --- | --- |
| Wellcome | 200848/Z/16/Z | David Ron |
| Wellcome | 996 100140 | David Ron |

The funders had no role in study design, data collection and interpretation, or the decision to submit the work for publication.

### Author contributions
Steffen Preissler, Conceptualization, Formal analysis, Investigation, Visualization, Methodology, Writing - original draft, Project administration; Claudia Rato, Formal analysis, Investigation, Methodology, Writing - review and editing; Yahui Yan, Data curation, Formal analysis, Investigation, Visualization, Writing - review and editing; Luke A Perera, Data curation, Formal analysis, Investigation, Writing - review and editing; Aron Czako, Investigation, Writing - review and editing; David Ron, Conceptualization, Funding acquisition, Methodology, Writing - original draft

### Author ORCIDs
Steffen Preissler  https://orcid.org/0000-0001-7936-9836
Claudia Rato  https://orcid.org/0000-0002-3971-046X
Yahui Yan  https://orcid.org/0000-0001-6934-9874
Luke A Perera  https://orcid.org/0000-0002-0032-1176
Aron Czako  https://orcid.org/0000-0002-3220-2083
David Ron  https://orcid.org/0000-0002-3014-5636

### Decision letter and Author response
Decision letter https://doi.org/10.7554/eLife.62601.sa1
Author response https://doi.org/10.7554/eLife.62601.sa2

## Additional files

### Supplementary files
• Supplementary file 1. Source data for the flow cytometry experiment shown in *Figure 1—figure supplement 2*.

• Supplementary file 2. Source data and calculated rates for MABA-ADP release experiment shown in *Figure 3—figure supplement 1*.

• Supplementary file 3. Source data and calculated rates for the MABA-ADP release experiment shown in *Figure 6—figure supplement 2A*.

• Supplementary file 4. Source data and calculated melting temperatures for the DSF experiment shown in *Figure 6—figure supplement 2B*.

• Supplementary file 5. Source data and calculated rates for the MABA-ADP release experiment shown in *Figure 6—figure supplement 3B*.

• Supplementary file 6. Source data for the SEC traces shown in *Figure 7—figure supplement 2*.

• Supplementary file 7. Table of plasmids.

• Transparent reporting form

### Data availability
Diffraction data have been deposited in PDB under the accession codes 6ZYH, 6ZYI, 6ZYJ, 7A4U, 7A4V.

The following datasets were generated:

| Author(s) | Year | Dataset title | Dataset URL | Database and Identifier |
|---|---|---|---|---|
| Yan Y, Preissler S, Ron D | 2020 | Crystal structure of GRP78 (70kDa heat shock protein 5 / BiP) ATPase domain in complex with ADP and calcium | https://www.rcsb.org/structure/6ZYH | RCSB Protein Data Bank, 6ZYH |
| Yan Y, Preissler S, Ron D | 2020 | Crystal structure of HSP70 ATPase domain in complex with ADP and | https://www.rcsb.org/structure/6ZYI | RCSB Protein Data Bank, 6ZYI |

| | | | calcium | | |
|---|---|---|---|---|---|
| Yan Y, Preissler S, Ron D | 2020 | Crystal structure of Hsc70 ATPase domain in complex with ADP and calcium | https://www.rcsb.org/structure/6ZYJ | RCSB Protein Data Bank, 6ZYJ |
| Perera LA, Ron D | 2020 | Crystal structure of lid-truncated apo BiP in an oligomeric state | https://www.rcsb.org/structure/7A4U | RCSB Protein Data Bank, 7A4U |
| Perera LA, Ron D | 2020 | Crystal structure of lid-truncated ADP-bound BiP in an oligomeric state | https://www.rcsb.org/structure/7A4V | RCSB Protein Data Bank, 7A4V |

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
