## [Decision Letter]

**Acceptance summary:**

Endoplasmic reticulum (ER) calcium depletion activates the unfolded protein response. Here the authors show that calcium selectively affects the dynamics of the chaperone BiP by enhancing its affinity for ADP. These findings explain the rapid dissociation of certain substrates from BiP in a calcium-depleted ER and suggest mechanisms for tuning ER quality control and coupling the unfolded protein response to signals that mobilize ER calcium in secretory cells.

**Decision letter after peer review:**

Thank you for submitting your article "Calcium depletion challenges endoplasmic reticulum proteostasis by destabilising BiP-substrate complexes" for consideration by *eLife*. We apologize sincerely for the delay in getting these comments to you. Your article has been reviewed by three peer reviewers, and the evaluation has been overseen by Suzanne Pfeffer as the Senior and Reviewing Editor. The reviewers have opted to remain anonymous.

The reviewers have discussed the reviews with one another and the Reviewing Editor has drafted this decision to help you prepare a revised submission.

Summary:

For many decades it was an intriguing observation that Ca^2+^ release from the endoplasmic reticulum (ER), either due to signalling processes or induced pharmacologically, triggered the unfolding protein response. However, the link between fluctuations of the Ca^2+^ concentrations in the ER and protein homeostasis remained unclear. Ron and co-workers investigated the molecular mechanism for this observation with respect to function of the Hsp70 chaperone BiP the main regulator of the unfolding protein response. They found that Ca^2+^ ion concentrations in the range of physiological fluctuations in the ER influences BiP-protein substrate complexes such that high Ca^2+^ levels reduce nucleotide exchange rates that are limiting for substrate release. This is a particular feature of metazoan ER Hsp70s and not found to a similar extent for yeast ER Hsp70, for eukaryotic cytosolic Hsp70s and for bacterial DnaK. The authors solved the crystal structure of the nucleotide binding domain of mammalian BiP, cytosolic constitutive Hsc70 and cytosolic heat inducible Hsp70 in complex with ADP and Ca^2+^ ions and found two Ca^2+^ ions bound in similar positions in all structures. Comparison of these structures with an earlier solved crystal structure of the nucleotide binding domain in complex with ADP and Mg^2+^ revealed differences that could explain the increased affinity of CaADP as compared to MgADP. Furthermore, by solving the crystal structure of oligomeric BiP and by isolating and characterizing oligomeric species, the authors elucidated the conundrum that BiP oligomers, though representing a substrate type interaction, was much more refractory to Ca^2+^-ion depletion than other BiP-substrate complexes. The experiments are conducted very carefully, and the conclusions drawn seem fully justified. Overall this is an excellent study that very elegantly solved a long-standing question in the Hsp70 field and the enigmatic link between Ca^2+^ signalling and proteostasis in the ER.

Because of the detailed and thoughtful comments of all the reviewers, their reviews are included for your benefit. With regard to what needs to be done, the reviewers provide this essential summary.

Essential revisions:

1) To validate the influence of the second Ca-ion in their crystal structure the authors should compare CaADP and MgADP binding to BiP-D257A and BiP-D257N.

2) To clarify the influence of the NEF Grp170 on substrate release in the presence and absence of Ca-ions the authors should repeat the BLI assay of Figure 4B conditions 3 and 4 +/- Grp170.

3) The structure of the paper might be more logical if it started with the biochemistry (there is a clear reason to pursue the effects of Ca^2+^ on BiP in view of the published record), and then finish with the UPR-relevant experiments, which leave more cell-biological loose ends.

The other major comments should be addressed by textual addenda.

Reviewer #1:

The crystal structure of the BiP-NBD with CaADP showed two Ca^2+^-ions bound to the NBD, one coordinated to α- and β-phosphate and a second coordinated by the carboxyl group of D257 and the backbone carbonyl of H252. The authors discard this observation, as there is no difference between the different Hsp70s. However, the way in which the second Ca^2+^ ion is coordinated is very peculiar as this is exactly at the hinge where subdomain IIB tilts outward upon binding of a nucleotide exchange factor. In the absence of NEFs the equivalent of D252 forms an electrostatic contact to the backbone carbonyl of the equivalent of H252, in the presence of a NEF this contact is broken. It is conceivable that this Ca^2+^ ion stabilizes this interaction. It was proposed earlier that the ATP binding cleft opens and closes also in the absence of nucleotide exchange factors to release nucleotide. Thus, the Ca^2+^ ion at this position could have a profound effect on nucleotide exchange rates. Such a hypothesis could be tested by mutating D257 to alanine or asparagine. The prediction would be that the effect of Ca^2+^on the exchange rate would be diminished. Such a mutant could be invaluable for further establishing the physiological role of the Ca^2+^ regulation.

Figure 1: the authors should state more clearly how the flow cytometry data are quantified. Was the geometric mean or the arithmetic mean determined? For logarithmically distributed data like fluorescence the geometric mean should be used. Also, were only the events within the gate considered? This should be clearly stated in the figure legend. TCRα seems to be much more expressed than the mutant TCRα-N/Q. Is the latter more subjected to ERAD. The authors should comment on the differences.

Reviewer #2:

The submitted manuscript aims to address how "fluctuations in ER calcium impact organellar proteostasis." The authors begin by reproducing the previously reported observation that depletion of ER Ca^2+^ results in a decreased association of BiP with the TCR α chain and increased TCR α chain secretion. Data obtained using a variety of assays with recombinant protein lead the authors to propose a mechanism for substrate release upon Ca^2+^ depletion, wherein calcium acts on the nucleotide exchange phase of the BiP catalytic cycle, slowing ADP dissociation from BiP and favoring ADP-to-ADP exchange, which prolongs association of BiP with peptide. A decrease in Ca^2+^ is inferred to enhance the effective rate of ATP-induced substrate release, decreasing BiP-substrate complexes. Structures of several Hsp70 NBD show Ca^2+^ associated with nucleotide, consistent with a role for Ca^2+^ in the ATPase cycle, versus an allosteric role. The authors argue that modulation of the ATPase cycle by calcium is unique to the metazoan BiP; higher levels of Ca^2+^ (likely beyond physiological fluctuations) are necessary to slow ADP release for DnaK or yeast BiP. The authors further speculate that increased BiP oligomerization upon Ca^2+^ depletion is a consequence of the slow dissociation of oligomers, allowing for a redistribution of BiP from BiP-substrate complexes to oligomers.

The manuscript presents a large amount of work, including a careful biochemical analysis of BiP. The data are convincing that Ca^2+^ (relative to Mg^2+^) alters the rates of nucleotide dissociation and association, which impacts the stability of BiP-substrate complexes. Importantly, the impact of Ca^2+^ in vitro falls within the anticipated range of intracellular calcium fluctuations, suggesting that the in vitro data are relevant to the in vivo secretion of substrate observed upon Tg treatment. Although not a major point in the manuscript, few full-length Hsp70 structures exist in the literature, and the presentation of an additional full-length BiP structure adds also to our knowledge base.

What remains less fully revealed by the presented work is whether the clean in vitro biochemical observations translate fully to the more complex environment of the ER.

1) It would be informative and strengthen the authors' conclusions if they further assessed how the presence of NEF (Grp170) impacts the differences in ADP, ATP, and peptide association/dissociation in the presence of Mg^2+^ and/or Ca^2+^. Are the differences in rates the authors observe in the presence of Mg^2+^ vs Ca^2+^ abrogated by the presence of NEF, which one expects will be present in the ER at the time of Ca^2+^ depletion? For example, how does the presence of Grp170 alter the relative BiP dissociation from substrate under the various Mg^2+^/Ca^2+^/ATP/ADP conditions measured in Figure 4B?

2) The authors make the interesting observation that increasing the load of folding protein in the ER (e.g. tunicamycin treatment, PERK inactivation) limits the secretion of TCR upon subsequence Ca^2+^ depletion. How do the authors explain this observation in light of their model? This observation from Figure 1 is never specifically revisited in the Discussion; however, the authors speculate that increased protein folding load will lower the lumenal ATP:ADP ratio. Are the authors suggesting that a change in ATP:ADP ratio may be the reason cells pretreated with tunicamycin are more recalcitrant to substrate release from BiP upon Ca^2+^ depletion? Is PERK inhibition expected to alter the BiP ATPase cycle / alter the ATP:ADP ratio?

3) It might be helpful if the Figure 8 model was split to depict the changes expected for a Ca^2+^ replete and Ca^2+^ depleted ER. As drawn the model implies a futile cycle role for Ca-ADP and a normal ATPase cycle for Mg-associated nucleotide. One presumes the authors are depicting a Ca^2+^ replete ER? Showing how the relative flux through the catalytic cycle changes upon depletion in a second scheme would help depict the overall manuscript model. Moreover, do the authors presume there is no Ca-ATP in a Ca^2+^ replete ER, or just that the Ca-ATP has little role due to a favored associated with Ca-ADP over Ca-ATP? As drawn, the model depicts oligomerization as influenced only by MgATP – in a calcium replete ER, is there a reason why MgATP (versus CaATP) would be the associated nucleotide?

Reviewer #3:

As I would expect from the Ron lab, there is nothing much to criticize about the quality of the data.

This manuscript addresses the role of Ca^2+^ in controlling functional properties of BiP. The earlier papers (notably Lodish and Kong) that recorded the remarkable effects of Ca^2+^ depletion on proteins that depend on the secretory pathway can now be interpreted in light of the findings reported here. These findings are supported by a massive amount of data including structural biological approaches and careful kinetic measurements. On the whole the data are complete and well-presented, although I might have ordered the various elements slightly differently: begin with the clear demonstration of the role of Ca^2+^ in controlling BiP function, then focus on the data now contained in Figure 1. In this reviewer's opinion, it might be better not to confound the issue of Ca^2+^ depletion with induction of a UPR at the very beginning of the Results section. The UPR-relevant aspect could be introduced perhaps at the very end of the dataset. The authors should make explicit that for the TCRa N/Q mutant, any impact of applying tunicamycin cannot be attributed to an effect on the reporter itself but must arise from perturbation of glycosylation of other ER-resident proteins, including unidentified proteins that might associate transiently with the reporter, as well as resulting from induction of a UPR. The following (short) section would benefit from a more direct explanation of the rationale for inclusion of tunicamycin: "Next, we examined whether reporter display on the plasma membrane could also arise from an increased burden of misfolded ER proteins, or if it is a feature specifically related to low ER Ca^2+^.We used TCRα-N/Q for these experiments due to its improved ER retention characteristics. Depletion of ER Ca^2+^ with the ionophore A23187 also led to accumulation of the reporter on the cell surface (Figure 1D)."

How the lack of the TCR partner subunit(s) , notably TCRb affect the overall structure and folding of TCRa should be discussed. The elimination off the 4 N-linked glycosylation sites further compromises the folding status of TCRa.

In my opinion the Introduction can be condensed, perhaps starting with the Kong Lodish and other papers that first reported the impact of Ca^2+^ depletion on secretion and the material discussed in paragraph five and further.

Results: “Ca^2+^ release-mediated reporter export.” Awkward.

---

## [Author Response]

Essential revisions:1) To validate the influence of the second Ca-ion in their crystal structure the authors should compare CaADP and MgADP binding to BiP-D257A and BiP-D257N.

We have introduced the D257A and D257N mutations into mammalian full-length BiP, purified the proteins and performed MABA-ADP release assays. The data are presented in new Figure 6—figure supplement 3B (together with the structural model highlighting the second Ca^2+^ binding site on BiP’s nucleotide binding domain in panel A). Both mutant proteins showed similar but slightly different basal MABA-ADP release rates compared to wild-type BiP. Importantly, however, like the wild-type protein, both BiP-D257A and BiP-D257N showed much slower MABA-ADP dissociation when nucleotide binding was performed in presence of CaCl_2_. Moreover, in all cases, presence of the nucleotide exchange factor Grp170 in the dissociation phase enhanced MABA-ADP release. These results indicate that the second Ca^2+^ binding site (observed in crystal structures of Hsp70 nucleotide binding domains) does not have a major contribution to the mechanism by which Ca^2+^ ions affect ADP dissociation from BiP that is studied in this paper.

2) To clarify the influence of the NEF Grp170 on substrate release in the presence and absence of Ca-ions the authors should repeat the BLI assay of Figure 4B conditions 3 and 4 +/- Grp170.

As suggested by the reviewers, we have performed the BLI experiments under the conditions 3 and 4 of Figure 4B and included nucleotide exchange factor (NEF) Grp170 during the BiP dissociation phase alongside ATP, ADP, MgCl2 -/+ CaCl_2_. The data are presented in the new panel Figure 4C. The basal BiP dissociation rates in absence and presence of CaCl_2_ were comparable to the ones observed in Figure 4B. Under both conditions, Grp170 accelerated BiP dissociation from substrate-loaded BLI sensors to similar extend (on average ~2.2- and ~2.5-fold without and with CaCl_2_, respectively). Therefore, the stabilising effect of Ca^2+^ on BiP-substrate interactions is maintained even when nucleotide exchange is stimulated by the presence of NEF. We discuss these results in the manuscript.

3) The structure of the paper might be more logical if it started with the biochemistry (there is a clear reason to pursue the effects of Ca2+ on BiP in view of the published record), and then finish with the UPR-relevant experiments, which leave more cell-biological loose ends.

We thank the reviewers for considering this editorial issue and for their suggested improvement. However, after examining the question empirically (in presentations of the data to colleagues) we feel that the narrative benefits from presenting the cell-based experiments first, because these make an important link to the previously-described, but yet unexplained, observations that hint to a significant influence of Ca^2+^ on BiP’s functionality and thus provide the conceptual background for the detailed mechanistic studies that follow.

Reviewer #1:The crystal structure of the BiP-NBD with CaADP showed two Ca^2+^-ions bound to the NBD, one coordinated to α- and β-phosphate and a second coordinated by the carboxyl group of D257 and the backbone carbonyl of H252. The authors discard this observation, as there is no difference between the different Hsp70s. However, the way in which the second Ca^2+^ ion is coordinated is very peculiar as this is exactly at the hinge where subdomain IIB tilts outward upon binding of a nucleotide exchange factor. In the absence of NEFs the equivalent of D252 forms an electrostatic contact to the backbone carbonyl of the equivalent of H252, in the presence of a NEF this contact is broken. It is conceivable that this Ca^2+^ ion stabilizes this interaction. It was proposed earlier that the ATP binding cleft opens and closes also in the absence of nucleotide exchange factors to release nucleotide. Thus, the Ca^2+^ ion at this position could have a profound effect on nucleotide exchange rates. Such a hypothesis could be tested by mutating D257 to alanine or asparagine. The prediction would be that the effect of Ca^2+^on the exchange rate would be diminished. Such a mutant could be invaluable for further establishing the physiological role of the Ca^2+^ regulation.

See response to Essential revisions point 1.

Figure 1: the authors should state more clearly how the flow cytometry data are quantified. Was the geometric mean or the arithmetic mean determined? For logarithmically distributed data like fluorescence the geometric mean should be used. Also, were only the events within the gate considered? This should be clearly stated in the figure legend. TCRα seems to be much more expressed than the mutant TCRα-N/Q. Is the latter more subjected to ERAD. The authors should comment on the differences.

In calculating the median fluorescent signal of any given population, all the cells were used (the horizonal and vertical lines on the 2D flow cytometry plot are guides, not borders of gates, as explained in the revised legend). The arithmetic mean of the median fluorescent signal of three experiments ± SD is plotted.

In the revised manuscript we comment on the differences in expression levels of the glycosylated and non-glycosylated versions of TCRα.

Reviewer #2:[…]1) It would be informative and strengthen the authors' conclusions if they further assessed how the presence of NEF (Grp170) impacts the differences in ADP, ATP, and peptide association/dissociation in the presence of Mg^2+^ and/or Ca^2+^. Are the differences in rates the authors observe in the presence of Mg^2+^ vs Ca^2+^ abrogated by the presence of NEF, which one expects will be present in the ER at the time of Ca^2+^ depletion? For example, how does the presence of Grp170 alter the relative BiP dissociation from substrate under the various Mg^2+^/Ca^2+^/ATP/ADP conditions measured in Figure 4B?

See Essential revisions point 2 outlined above.

2) The authors make the interesting observation that increasing the load of folding protein in the ER (e.g. tunicamycin treatment, PERK inactivation) limits the secretion of TCR upon subsequence Ca^2+^ depletion. How do the authors explain this observation in light of their model? This observation from Figure 1 is never specifically revisited in the Discussion; however, the authors speculate that increased protein folding load will lower the lumenal ATP:ADP ratio. Are the authors suggesting that a change in ATP:ADP ratio may be the reason cells pretreated with tunicamycin are more recalcitrant to substrate release from BiP upon Ca^2+^ depletion? Is PERK inhibition expected to alter the BiP ATPase cycle / alter the ATP:ADP ratio?

We agree with the reviewer that there might be a link between unfolded protein stress (caused e.g. by tunicamycin treatment) and a decrease in the ATP:ADP ratio, which may counter the effects of ER Ca^2+^ depletion on BiP, such as enhanced export of substrate proteins like the TCRα reporter. To make this point clearer, we have indicated the predicted effect of a change in the ER ATP:ADP ratio on BiP’s chaperone cycle in the new model presented in Figure 8A. See also explanation in the Discussion section.

3) It might be helpful if the Figure 8 model was split to depict the changes expected for a Ca^2+^ replete and Ca^2+^ depleted ER. As drawn the model implies a futile cycle role for Ca-ADP and a normal ATPase cycle for Mg-associated nucleotide. One presumes the authors are depicting a Ca^2+^ replete ER? Showing how the relative flux through the catalytic cycle changes upon depletion in a second scheme would help depict the overall manuscript model. Moreover, do the authors presume there is no Ca-ATP in a Ca^2+^ replete ER, or just that the Ca-ATP has little role due to a favored associated with Ca-ADP over Ca-ATP? As drawn, the model depicts oligomerization as influenced only by MgATP – in a calcium replete ER, is there a reason why MgATP (versus CaATP) would be the associated nucleotide?

We thank the reviewer for pointing out the shortcomings of the original model. We have generated new models for Figure 8: Panel A is a refined version of the original model depicting BiP’s chaperone cycle and substrate interaction-stabilising ADP-to-ADP exchange cycles. We now also indicate the influence of changes in both the ER Ca^2+^ levels and the ATP:ADP ratio on nucleotide exchange. To further increase clarity, we present in panel B the distribution of BiP amongst different pools in a Ca^2+^-replete ER and its observed relative redistribution in a Ca^2+^-depleted ER as separate cartoons.

Reviewer #3:As I would expect from the Ron lab, there is nothing much to criticize about the quality of the data.This manuscript addresses the role of Ca2+ in controlling functional properties of BiP. The earlier papers (notably Lodish and Kong) that recorded the remarkable effects of Ca2+ depletion on proteins that depend on the secretory pathway can now be interpreted in light of the findings reported here. These findings are supported by a massive amount of data including structural biological approaches and careful kinetic measurements. On the whole the data are complete and well-presented, although I might have ordered the various elements slightly differently: begin with the clear demonstration of the role of Ca2+ in controlling BiP function, then focus on the data now contained in Figure 1. In this reviewer's opinion, it might be better not to confound the issue of Ca2+ depletion with induction of a UPR at the very beginning of the Results section. The UPR-relevant aspect could be introduced perhaps at the very end of the dataset. The authors should make explicit that for the TCRa N/Q mutant, any impact of applying tunicamycin cannot be attributed to an effect on the reporter itself but must arise from perturbation of glycosylation of other ER-resident proteins, including unidentified proteins that might associate transiently with the reporter, as well as resulting from induction of a UPR. The following (short) section would benefit from a more direct explanation of the rationale for inclusion of tunicamycin: "Next, we examined whether reporter display on the plasma membrane could also arise from an increased burden of misfolded ER proteins, or if it is a feature specifically related to low ER Ca^2+^.We used TCRα-N/Q for these experiments due to its improved ER retention characteristics. Depletion of ER Ca^2+^ with the ionophore A23187 also led to accumulation of the reporter on the cell surface (Figure 1D)."

In the revised version we clarify the utility of the unglycosylated version of TCRa to the experiments involving the use of tunicamycin. We thank the reviewer for drawing our attention to this point.

How the lack of the TCR partner subunit(s) , notably TCRb affect the overall structure and folding of TCRa should be discussed. The elimination off the 4 N-linked glycosylation sites further compromises the folding status of TCRa.

In the revised version we emphasise that impact of expression of TCRα as an orphan and its importance to the experimental design.

In my opinion the Introduction can be condensed, perhaps starting with the Kong Lodish and other papers that first reported the impact of Ca2+ depletion on secretion and the material discussed in paragraph five and further.

We have shortened the front end of the Introduction, as suggested by the reviewer, focussing on calcium.

Results: “Ca^2+^ release-mediated reporter export.” Awkward.

The sentence has been changed.